

# A constraint upon the basal water distribution and basal thermal state of the Greenland Ice Sheet from radar bed-echoes

Thomas M. Jordan[1,2], Christopher N. Williams[1,3], Dustin. M. Schroeder[2,4], Yasmina M. Martos[5,6], Michael A. Cooper[1], Martin. J. Siegert[7], John D. Paden[8], Phillipe Huybrechts[9], and Jonathan L. Bamber[1]

[1]Bristol Glaciology Centre, School of Geographical Sciences, University of Bristol, Bristol, UK.
[2]Department of Geophysics, Stanford University, Stanford, CA, USA.
[3]Now at British Geological Survey, Nottingham, UK.
[4]Department of Electrical Engineering, Stanford University, Stanford, CA, USA.
[5]Department of Astronomy, University of Maryland, College Park, MD, USA.
[6]NASA Goddard Space Flight Center, Greenbelt, MD, USA.
[7]Grantham Institute and Department of Earth Science and Engineering, Imperial College, London, UK.
[8]Center for Remote Sensing of Ice Sheets, University of Kansas, Lawrence, USA.
[9]Earth System Science and Departement Geografie, Vrije Universiteit Brussel, Brussels, Belgium.

**Abstract.** There is widespread, but often indirect, evidence that a significant fraction of the bed beneath the Greenland Ice Sheet is thawed (at or above the pressure melting point for ice). This includes the beds of major outlet glaciers and their tributaries and a large area around the North-GRIP borehole in the ice-sheet interior. The ice-sheet scale distribution of basal water is, however,

poorly constrained by existing observations. In principle, airborne radio-echo sounding (RES) enables the detection of basal water from bed-echo reflectivity, but unambiguous mapping is limited by uncertainty in signal attenuation. Here we introduce a new RES diagnostic for basal water that is associated with wet to dry transitions in bed material: bed-echo reflectivity variability. Importantly, this diagnostic is demonstrated to be attenuation-insensitive and the technique enables combined

analysis of over a decade of Operation IceBridge survey data.

The basal water predictions are compared with existing analyses for the basal thermal state (frozen and thawed beds) and geothermal heat flux. In addition to the outlet glaciers, we demonstrate widespread water storage in the northern and eastern interior. Notably, we observe a quasi-linear 'corridor' of basal water extending from NorthGRIP to Petermann glacier that spatially correlates with elevated heat flux predicted by a recent magnetic model. Finally, with a general aim to stim-

ulate regional and process specific investigations, the basal water predictions are compared with bed topography, subglacial flow paths, and ice-sheet motion. The basal water distribution, and its relationship with the basal thermal state, provides a new constraint for numerical models.



## 1 Introduction

Basal water beneath the Greenland Ice Sheet (GrIS) influences, and is influenced by, the dynamics
and thermodynamics of the overlying ice. A lubricated bed is a necessary condition for basal sliding,
which can be responsible for up to about 90% of the ice surface velocity (van der Veen, 2013).
Constraining the spatial distribution of basal water is important, therefore, for understanding the
dynamic state of the overlying ice and its sensitivity to external forcing. A reliable estimate of the

presence of basal water can also be used as a boundary condition/constraint in numerical modelling
and to evaluate model performance and is, as a consequence, an attractive objective.

The spatial distribution of basal water beneath the GrIS is known to arise from an interplay of
different physical processes including: surface melt (e.g. van de Wal et al. (2008)), basal melting
due to geothermal heat (e.g. Dahl-Jensen et al. (2003); Rogozhina et al. (2016)), frictional and shear

heating (e.g. van der Veen (2013)), and transport processes (surface, englacial and subglacial) which
redistribute water (e.g. Rennermalm et al. (2013); Chu (2014)). There are, however, limited ob-
servational constraints on the ice-sheet scale distribution of basal water, and the relationship with
other glacial and subglacial properties is therefore largely unexplored and/or speculative. The lack
of unambiguous information about basal water arises primarily because there are only a few existing

observations of subglacial lakes (Palmer et al., 2013; Howat et al., 2015; Willis et al., 2015; Palmer
et al., 2015). By contrast, the Antarctic Ice Sheet currently has over 400 identified subglacial lakes
(Siegert et al., 2016)) some of which have been found to be actively draining/recharging. Instead,
there is evidence that basal water beneath the GrIS exists in smaller pools (Chu et al., 2016), as wet
sediment (Christianson et al., 2014), and as part of channelised networks (Livingstone et al., 2017).

The basal temperature distribution of the GrIS determines where basal water can exist, and re-
quires basal temperatures at or above the pressure melting point (PMP) for ice (from herein 'thawed').
Direct basal temperature measurements are, however, sparse. At the majority of the interior bore-
holes - Camp Century, Dye 3, GRIP, GISP2, and NEEM - basal temperatures indicate frozen beds
(Weertman, 1968; Gundestrup and Hansen, 1984; Dahl-Jensen et al., 1998; Cuffey et al., 1995; Mac-

Gregor et al., 2016), with the thawed bed at NorthGRIP an exception (Andersen et al., 2004). Toward
the ice-sheet margins boreholes generally indicate thawed beds (MacGregor et al., 2016). Indirect
methodologies for discriminating frozen and thawed beds (ice-sheet model predictions, radiostratig-
raphy, and surface properties) were recently combined by MacGregor et al. (2016), to produce a
frozen-thawed likelihood map for beds beneath the GrIS. Key predictions were that the central ice

divides and west facing slopes generally have frozen beds, the southern and western outlet glaciers
have a thawed bed, and that basal thaw extends east from NorthGRIP over a large fraction of the
northeastern ice sheet.

Spatially variable geothermal heat flux (GHF) influence the basal temperature distribution (Dahl-
Jensen et al., 2003; Greve, 2005; Rogozhina et al., 2016), hydrology (Rogozhina et al., 2016), and

flow features (Fahnestock et al., 2001) in the interior of the GrIS. Notably, the onset of the North




East Greenland Ice Stream (NEGIS) is predicted to arise from basal melting that is attributed to locally elevated GHF (Fahnestock et al., 2001), which, in turn, has recently been attributed to the path of the Iceland hotspot track (Rogozhina et al., 2016; Martos et al., In revision). As with basal temperature, the sparsity of borehole measurements limits direct inference of GHF (which is related to the vertical gradient of basal temperature). Instead, a range of geophysical techniques including: tectonic (Pollack et al., 1993), seismic (Shapiro and Ritzwoller, 2004), and magnetic (Fox Maule et al., 2009; Martos et al., In revision) models have been used to map GHF beneath the ice sheet. These models, however, differ greatly in the predicted spatial distribution for GHF and also in the absolute values (Rogozhina et al., 2012). Due to the relationship between basal melt and GHF, basal water observations can be used to further refine and cross-validate GHF models (Schroeder et al., 2014; Rogozhina et al., 2016).

In principle, airborne radio-echo sounding (RES) surveys provide the information and spatial coverage to infer the presence of basal water at the ice-sheet scale. Bed-echo reflectivity is the most commonly used diagnostic for this purpose (e.g. Peters et al. (2005); Jacobel et al. (2009); Oswald and Gogineni (2008)), and is based upon the prediction that, across a range of subglacial materials, wet glacier beds have a higher reflectivity than dry or frozen beds (Bogorodsky et al., 1983; Martinez et al., 2001; Peters et al., 2005). However, due to uncertainty and spatial variation in radar attenuation (an exponential function of temperature (Corr et al., 1993)) bed-echo reflectivity is subject to spatial bias and can be ambiguous when mapped over larger regions (Matsuoka, 2011; MacGregor et al., 2012; Jordan et al., 2016). In order to mitigate spatial bias from radar attenuation, bed-echo scattering properties - including the 'specularity' (a measure of the angular distribution of energy and, consequently, the smoothness of the bed at a radar-scale wavelength) (Schroeder et al., 2013; Young et al., 2016) and the bed-echo 'abruptness' (a waveform parameter) (Oswald and Gogineni, 2008, 2012) - have also been employed in basal water detection. Although attenuation-independent, use of some bed-echo scattering properties to discriminate basal water can be prone to ambiguity and can potentially lead to false-positive detection of smoother bedrock as water (Jordan et al., 2017).

In this study we introduce a new RES diagnostic for basal water which is specifically tuned to detect wet to dry transitions: bed-echo reflectivity variability. This RES diagnostic is demonstrated to be highly insensitive to radar attenuation, thus reducing the likelihood of false-positive identification of basal water. Moreover, it also only requires local radiometric power calibration, and thus enables different Operation IceBridge RES campaigns, using different radar system settings (e.g. peak transmit power, antenna gain), to be combined and mapped. Whilst only encompassing a subset of basal water (specifically, finite water bodies with sharp horizontal gradients in water content) the RES diagnostic enables a new ice-sheet scale constraint to be placed upon where the bed of the GrIS is thawed. The primary focus of the study is therefore to compare the spatial relationship between predicted basal water and up-to-date analyses for the basal thermal state (MacGregor et al.,



2016) and GHF (Shapiro and Ritzwoller, 2004; Fox Maule et al., 2009; Martos et al., In revision) beneath the GrIS. We observe new predictions for basal water predominantly in the northern and

eastern ice sheet, which spatially correlate with elevated GHF recently inferred by Martos et al. (In revision). We then compare basal water and bed topography (Morlighem et al., 2017), which enables us to identify likely subglacial flow paths and storage locations beneath the contemporary ice sheet. Finally, we compare the relationship between basal water and ice surface speed (Joughin et al., 2010, 2016).

## 2   Methods

### 2.1   Radio-echo sounding data and bed-echo analysis

The airborne RES data used in this study were collected by the Center for Remote Sensing of Ice Sheets (CReSIS) over the time period from 2003-2014. The data were taken using a succession of radar instruments: Advanced Coherent Radar Depth Sounder (ACORDS), Multi-Channel

Radar Depth Sounder (MCRDS), Multi-Channel Coherent Radar Depth Sounder (MCoRDS), Multi-Channel Coherent Radar Depth Sounder: Version 2 (MCoRDS v2), mounted on three airborne platforms: P-3B Orion (P3), DHC-6 Twin Otter (TO) DC8, Douglas DC-8 (DC8) (Paden, 2015). The flight-track coverage, subdivided by radar system, is shown in Fig. 1(a) with the season breakdown: ACORDS 2003 P3 and 2005 TO; MCRDS 2006 TO, 2007 P3, 2008 TO and 2009 TO; MCoRDs

2010 P3 and 2010 DC8; MCoRDs v2 2011 TO, 2011 P3, 2012 P3, 2013 P3, 2014 P3. More details on the track lengths and data segmentation can be found in MacGregor et al. (2015a). The vast majority of the data were collected in the months March-May.

The various radar system details and signal processing steps are described in detail in previous works (Rodriguez-Morales et al., 2014; Gogineni et al., 2014; Paden, 2015). The centre-frequency of

the radar systems is either 150 MHz (ACORDS and MCRDS) or 195 MHz (MCoRDs and MCoRDS v2), and, after accounting for pulse-shaping and windowing, the depth-range (vertical) resolution can vary from $\sim$ 4.3-20 m in ice. The along-track (horizontal) resolution also varies between field seasons, and is typically $\sim$ 30 or 60 m. The radar-echo strength profiles (Level 1B data) employ fixed fast-time gain where the receiver gain is constant for each recorded pulse which enables consis-

tent interpretation of bed-echo power on a season-by-season basis. Whilst transmitted power can differ between seasons, since we consider local variability, offsets between seasons do not matter which enables straightforward inter-season data combination. Pre-2003 CReSIS data uses manual gain control and hence these seasons are not included.

The procedure to extract bed-echo power is similar to Oswald and Gogineni (2008); Jordan et al.

(2016, 2017) and aggregates power over bed-echo fading (i.e. performs a depth-range integral). Briefly, the procedure consists of the following steps. Firstly, CReSIS ice thickness (Level 2) picks are used as initial estimate for the depth-range bin of the peak bed-echo power. Secondly, a local





re-tracker is used to locate the true depth-range bin for peak bed-echo power. Thirdly, the power is aggregated by a discrete summation over the bed-echo envelope (both before and after the peak). The summations are truncated when the power is 10 dB or less than the peak, thus to try to ensure that the integral consists of a dominant peak associated with a dominant reflecting facet. Finally, to ensure a suitable signal-to-noise ratio, a 'quality control' measure is imposed such that bed-echo peak power must be 10 dB above the noise floor. This results in the effective coverage shown in Fig. 1(b). Regions with 'poor' quality bed-echoes include a spatially coherent coverage gap in the southern interior, high altitude data, and some marginal regions.

The rationale for use of aggregated bed-echo power (over peak power) is that it serves to reduce bed-echo power variability due to roughness, and thus better enables comparison with the specular bed-echo reflectivity values that are used to infer bulk material properties (Oswald and Gogineni, 2008). Additionally, since roughness scattering loss is frequency-dependent (MacGregor et al., 2013), aggregated power serves as a pragmatic way to best combine bed-echo power measurements from the 195 MHz and 150 MHz radar systems. This is supported by the observed $\sim 1$ dB greater scattering loss (estimated here by the dB difference between peak and aggregated power) for the 195 MHz systems at cross-over points.

Key landmarks of the GrIS that are used in the data description - temperature boreholes, drainage basin boundaries, and major fast flow regions - are shown in Fig. 1(c).

## 2.2 Bed-echo power, attenuation correction and bed-echo reflectivity

The bulk material properties of glacier beds, including the presence of basal water, can, in principle, be inferred from the reflectivity of the bed-echo (Bogorodsky et al., 1983; Peters et al., 2005). The reflectivity, $[R]$, is obtained from solving the decibel form of the bed-echo power equation

$$[P] = [S] - [G] + [R] - [L] - [B], \tag{1}$$

where $[P]$ is the bed-echo power (in this case the aggregated bed-echo power), $[S]$ is the system performance, $[G]$ is the geometric correction, $[L]$ is the attenuation loss in ice, and $[B]$ is the loss due to birefringence (ice fabric anisotropy), and the notation $[X] = 10\log_{10} X$ is assumed (Matsuoka et al., 2010). The geometric correction for a specular reflector can be defined by

$$[G] = 2[2(s + h/\sqrt{\epsilon_{ice}})], \tag{2}$$

where $s$ is the aircraft height and $h$ is the ice thickness and $\epsilon_{ice} = 3.15$ is the relative dielectric permittivity of ice to give the geometrically-corrected bed-echo power

$$[P_g] = [P] + [G]. \tag{3}$$

(e.g. Schroeder et al. (2016a)). For the majority of the CReSIS data used (2006 TO onward) $s$ and $h$ are known and eq. (2) can be applied exactly. For the 2003 P3 and 2005 TO seasons only $h$ is known



and $s = 500$ m is assumed (approximately the mean aircraft height). This approach is justifiable since in this study we are interested in local (length scale $\sim 5$ km) power variation, where $s$ varies slowly. It is assumed that variation in $[S]$ and $[B]$ is also negligible (again, approximations that are strengthened by consideration of local power variation) then eq. (1) reduces to

$$[P_g] = [R] - [L], \tag{4}$$

where $[P_g] = [P] + [G]$ is geometrically-corrected bed-echo power. Finally, setting $[L] = 2 < N > h$ gives

$$[P_g] = [R] - 2 < N > h, \tag{5}$$

where $< N >$ (dB km$^{-1}$) is the (one-way) depth-averaged attenuation rate (Matsuoka et al., 2010).

A fundamental ambiguity in bed-echo reflectivity analysis is that there are two unknowns in eq. (5): $< N >$ and $[R]$. Two approaches are typically used to determine $< N >$: (i) 'forward modelling' using estimates of attenuation as a function of englacial temperature (e.g. MacGregor et al. (2007); Matsuoka et al. (2012b); Chu et al. (2016)), (ii) 'empirical-determination' using the linear regression of bed-echo power and ice thickness (e.g. Jacobel et al. (2009); Schroeder et al. (2016b)).

Attenuation follows an Arrhenius (exponential) relationship with temperature and a linear dependence upon the concentration of ionic impurities: primarily hydrogen (H$^+$), but also chlorine (Cl$^-$), and ammonium (NH$_4^+$) (Corr et al., 1993; Wolff et al., 1997; MacGregor et al., 2007, 2015b). On an ice-sheet scale scale, the uncertainty when 'forward' modelling $< N >$ is so high that it can be prohibitively challenging to accurately calibrate $[R]$ (Matsuoka et al., 2012a; MacGregor et al.,

2015b; Jordan et al., 2016). This is due to both uncertainty in ice-sheet model temperature fields, the ionic concentrations, and the tuning of the parameters in the Arrhenius equation (MacGregor et al., 2007, 2015b). Empirical determination of $< N >$ using bed-echo power is also subject to sources of potential bias. In particular, the regression methods can be ill-posed when there is rapid spatial variation in attenuation (Matsuoka et al., 2012a), or when there is a thickness-correlated distribution

in bed-echo reflectivity (Jordan et al., 2016).

    We will later demonstrate that, unlike absolute values of $[R]$, local variability in bed-echo reflectivity is highly insensitive to modelled values of $< N >$ (Sect. 2.6). However, despite acting as a very weak constraint, an initial estimate for ice-sheet scale variation in $< N >$ is still necessary to calculate reflectivity variability. The estimate for $< N >$ relies on previous work by Jordan et al.

(2016) and uses the 'M07' Arrhenius equation MacGregor et al. (2007), the Greenland Ice Sheet Model (GISM) temperature field from Huybrechts (1996) as updated in Goelzer et al. (2013), depth-averaged ionic concentrations from the GRIP ice core (MacGregor et al., 2015b), and the Greenland ice thickness data set in Bamber et al. (2013a). Using this model framework, it is that predicted that $< N >$ varies by a factor $\sim 5$ over the GrIS, ranging from $\sim 6$ dB km$^{-1}$ in the colder northern

interior to $\sim 30$ dB km$^{-1}$ toward the warmer southwestern margins (refer to Fig. 5(a) in Jordan et al. (2016) for a spatial plot).





### 2.3 Calculating bed-echo power and reflectivity variability

When calculating bed-echo power and reflectivity variability from the along-track data we chose
to work with the standard deviations, $\sigma_{[P_g]}$ and $\sigma_{[R]}$, which have familiar units of dB. Since this
is a non-standard approach to bed-echo data analysis, we now take a closer look at the statistical
properties. The formula for $\sigma_{[P_g]}$ follows from the variance of eq. (4) and is given by

$$\sigma_{[P_g]} = \sqrt{\sigma_{[R]}^2 + \sigma_{[L]}^2 - 2\sigma_{[R],[L]}}, \tag{6}$$

where $\sigma_{[L]}$ is the standard deviation in attenuation loss, and $\sigma_{[R],[L]}$ is the covariance of bed-echo
reflectivity and attenuation loss. Using $[L] = 2 < N > h$ and assuming $< N >$ can be approximated
as constant (justifiable at the 5 km length scale that is later considered) then eq. (6) becomes

$$\sigma_{[P_g]} = \sqrt{\sigma_{[R]}^2 + 4 < N >^2 \sigma_h^2 - 4 < N > \sigma_{[R],h}}, \tag{7}$$

where $\sigma_h$ is the standard deviation of ice thickness and $\sigma_{[R],h}$ is the covariance of bed-echo reflectiv-
ity and ice thickness. In regions where $\sigma_{[R],[L]} \approx 0$, (bed-echo reflectivity has negligible covariance
with attenuation loss), eq. (6) and eq. (7) are approximated by

$$
\begin{aligned}
\quad \sigma_{[P_g]} \quad &\approx \quad \sqrt{\sigma_{[R]}^2 + \sigma_{[L]}^2}, &\tag{8}\\
&\approx \quad \sqrt{\sigma_{[R]}^2 + 4 < N >^2 \sigma_h^2}, &\tag{9}
\end{aligned}
$$

and the loss component of $\sigma_{[P_g]}$ is solely modulated by $\sigma_{[L]}$ which is proportional to the product
$< N > \sigma_h$. Whilst an approximation (in certain circumstances the second and third terms on the
right hand side of eq. (6) and eq. (7) can be of comparable magnitude) this scenario provides an
intuitive way to understand the interrelationship between $\sigma_{[P_g]}$, $\sigma_{[R]}$ and $\sigma_{[L]}$.

The numerical calculation of $\sigma_{[P_g]}$ and $\sigma_{[R]}$ is analogous to how topographic roughness (the rms
height) is calculated from bed elevation profiles (Shepard et al., 2001). In the calculations an along-
track 'window' of length 5 km at 1 km intervals was assumed and two example profiles for $[P_g]$,
$\sigma_{[P_g]}$, $[R]$, $\sigma_{[R]}$, $[L]$, $\sigma_{[L]}$ and $h$ are shown in Fig. 2. Fig. 2(a) is a representative example from
the interior of the ice sheet where $\sigma_{[L]}$ is relatively low and $h$ is thick ($\sim 2.8$ km). Subsequently
the profiles for $\sigma_{[P_g]}$ and $\sigma_{[R]}$ are very similar in appearance, with the most notable difference at
distance $\sim 360$ km where there is higher $\sigma_{[P_g]}$ due to the subglacial trough. The peaks in $\sigma_{[R]}$ are
later related to wet to dry bed material transitions in Sect. 2.5. It is also important to note that $\sigma_{[R]}$
can be greater than $\sigma_{[P_g]}$ (e.g. at distance $\sim 342$ km), which is an effect that can be explained by the
covariance between attenuation loss/ice thickness and bed-echo reflectivity in eq. (6) and eq. (7)).
Fig. 2(b) is a representative example from toward the ice-sheet margins where $\sigma_{[L]}$ is higher due to
more rapid variation in $h$ (more complex bed topography) and higher values of $< N >$ (warmer ice).
In this case, $\sigma_{[P_g]}$ is noticeably greater than $\sigma_{[R]}$, with the differences largely attributable to higher
$\sigma_{[L]}$ as anticipated by eq. (8).





When calculating $\sigma_{[P_g]}$, $\sigma_{[R]}$ and $\sigma_{[L]}$, bed-echo coverage gaps within a 5 km bin (see Fig. 1(b))
were accounted for by neglecting bins where less than half the data corresponded to 'good' bed-
echoes. The effects of this filtering step are demonstrated in Fig. 2(b) where, aligned with the deep
subglacial trough at distance $\sim 1324$ km, there are along-track gaps in $\sigma_{[P_g]}$, $\sigma_{[R]}$ and $\sigma_{[L]}$.

The 5 km length scale in the variability calculations was initially chosen for consistency with
the basal thermal state mask in MacGregor et al. (2016), which was deemed an appropriate scale
for integration of radar data with thermomechanical models at the ice-sheet scale. Both examples
in Fig. 2 highlight that, at this length scale, higher values of $\sigma_{[R]}$ can arise due to either a large
single transition in $[R]$, multiple smaller transitions/fluctuations, or a combination of both signal
components. The physical consequences and interpretation of the 5 km length scale in the context of
water detection is discussed in Sect. 2.5.

### 2.4    Distributions for bed-echo power and reflectivity variability

Spatial distributions for the variability measures: $\sigma_{[P_g]}$, $\sigma_{[L]}$, $\sigma_{[R]}$ are shown in Fig. 3(a,b,c). To aid
the, interpretation ice thickness (Morlighem et al., 2017) is shown in Fig. 3(d). In general, $\sigma_{[P_g]}$ has
a strong ice thickness dependence, and increases toward the margins where ice is thinner. The atten-
uation correction, which primarily acts to reduce the component of $\sigma_{[P_g]}$ that is attributable to $\sigma_{[L]}$,
results in a more uniform ice-sheet scale distribution of $\sigma_{[R]}$ than $\sigma_{[P_g]}$. Notably, there are localised
patches of higher $\sigma_{[R]}$ present in both marginal and interior regions (which are later attributed to the
presence of basal water). The ice-sheet scale trends in $\sigma_{[P_g]}$ and $\sigma_{[L]} = 2 < N > \sigma_h$ can be related to
spatial variation in $< N >$ (MacGregor et al., 2015b; Jordan et al., 2016) and bed roughness (Rippin,
2013; Jordan et al., 2017) (which correlates with $\sigma_h$) .

The two zoom regions in Fig. 3 include the flight-track profiles in Fig. 2. (north-central ice sheet,
pink bounding box; northwestern margins, black bounding box). These examples serve to further
illustrate the spatial interrelationship between $\sigma_{[P_g]}$, $\sigma_{[R]}$ and $\sigma_{[L]}$ in a typical interior region with
lower $\sigma_{[L]}$ and a typical marginal region with higher $\sigma_{[L]}$. Its is clear that the interior example has
very similar spatial distributions for $\sigma_{[P_g]}$ and $\sigma_{[R]}$, whereas the marginal example has higher $\sigma_{[P_g]}$
associated with the higher $\sigma_{[L]}$ that occurs in the subglacial troughs and more complex topography
toward the edge of the ice sheet. The marginal example also demonstrates that the power variability
associated with the subglacial troughs is largely removed for $\sigma_{[R]}$.

The corresponding frequency distributions for $\sigma_{[P_g]}$ and $\sigma_{[R]}$ are shown in Fig. 4. Both demon-
strate a strong positive-skew, with a long-tail extending to higher values. The mean and standard
deviation for $\sigma_{[P_g]}$ is greater than $\sigma_{[R]}$. This is consistent with the commonly observed result that
making an attenuation correction to $[P_g]$ acts to reduce the overall decibel range for $[R]$, (e.g. Os-
wald and Gogineni (2008); Schroeder et al. (2016b)), hence more closely resembling the predicted
dB range for bed materials (Peters et al., 2005).





### 2.5 Interpretation of reflectivity variability as a diagnostic for water

Radar bed-echo reflectivity depends upon the dielectric contrast between glacier ice and bed material. For a specular, nadir reflection the Fresnel power reflection coefficient is given by

$$[R] = 10 \log_{10} \left| \frac{\sqrt{\widetilde{\epsilon_{bed}}} - \sqrt{\widetilde{\epsilon_{ice}}}}{\sqrt{\widetilde{\epsilon_{bed}}} + \sqrt{\widetilde{\epsilon_{ice}}}} \right|^2, \tag{10}$$

where $\widetilde{\epsilon_{ice}}$ and $\widetilde{\epsilon_{bed}}$ are the complex dielectric permittivitiy of the glacier ice and bed material respectively. The relative (real) part of the permittivity, $\epsilon_{bed}$, is the primary control upon $[R]$. A summary of dielectric and reflective properties of glacier bed materials at typical ice-penetrating radar frequencies is given in Table 1 and is collated from Bogorodsky et al. (1983); Martinez et al. (2001); Peters et al. (2005). The permittivity and reflectivity range for each material arises due to sub-wavelength dielectric mixing between either ice or water and the bed material, and takes into account typical saturation and porosity values (Martinez et al., 2001; Peters et al., 2005). In general, lower values of $\epsilon_{bed}$ and $[R]$ occur for dry or frozen bed materials (approximately $\epsilon_{bed} < 7$ and $[R] < $ -14 dB), whilst higher values occur for wet bed materials (approximately $\epsilon_{bed} > 7$ and $[R] > $ -14 dB). Dielectric mixing between bed materials can also occur at the length scale of the Fresnel zone ($\sim$ 100 m), which results in a further averaging of the observed reflectivity (Berry, 1975; Peters et al., 2005).

Due to the range of possible bed materials at the ice-sheet scale it is not possible to formulate a unique dielectric model for diagnosing water from $\sigma_{[R]}$. A simple 'two-state' dielectric model, does, however, enable us to physically motivate the water diagnostic in terms of dielectric properties (Fig. 5). The model assumes that the along-track sample window is comprised of two different bed materials: the dry 'background' bed material with permittivity $\epsilon_{dry}$ and reflectivity $[R_{dry}]$, and the wet material with permittivity $\epsilon_{wet}$ and reflectivity $[R_{wet}]$. For simplicity, it is assumed that each along-track measurement is in one of the wet or dry states, with the wet-dry mixing ratio parameterised by $f$. In this formulation, a single body of wet material or multiple smaller bodies of wet material, have the same formula for the reflectivity variability given by

$$\sigma_{[R]} = \Delta[R] \sqrt{f^2(1-f) + (1-f)^2 f}, \tag{11}$$

where $\Delta[R] = [R_{wet}] - [R_{dry}]$ is the reflectivity difference between wet and dry beds. Eq. (11) is derived by considering the weighted variance for two discrete random variables and does not account for non-linear variations due to variable scattering coherence. A phase-space plot for $\sigma_{[R]}(f, \Delta[R])$ is shown in Fig. 5(c), and shows that for fixed $\Delta[R]$, $\sigma_{[R]}$ is maximised when $f = 0.5$ (i.e. an even mixing of wet and dry materials).

Past diagnosis of basal water typically associates the upper tail of the reflectivity distribution with water, prescribing a threshold above which the bed is interpreted as wet (e.g. Jacobel et al. (2009); Chu et al. (2016)). In this study, a similar thresholding approach is applied to the distribution of $\sigma_{[R]}$ (Fig. 4(b)). The threshold choice for basal water ($\sigma_{[R]} > 6$ dB) corresponds to the region greater than





the $\sigma_{[R]}$ = 6 dB contour in Fig. 5(c) and requires a minimum wet to dry reflectivity difference of
$\Delta[R]$ > 12 dB. In general, $\Delta[R]$ > 12 dB is only possible for a mixture of wet and dry (or frozen)
bed materials (Table 1). For example, an even mixing of ground water and dry granite ($f$ = 0.5,
$\Delta[R]$ = 17 dB) has $\sigma_{[R]}$ = 8.5 dB. The contours in Fig. 5(c) demonstrate that small perturbations
to even mixing (f $\neq$ 0.5) produce similar $\sigma_{[R]}$, and hence that water detection is insensitive to the
discretisation of the along-track sample window (Sect. 2.3). Overall, the threshold choice ($\sigma_{[R]}$ >
6 dB) is fairly conservative and is deliberately intended to reduce false-positive detection of basal
water (at the expense of reduced overall detection).

The bed-echo power aggregation in Sect. 2.1 partially mitigates for roughness-induced scattering
loss and the along-track power variability associated with this. Additionally, we later demonstrate
that the water detection method/semi-empirical threshold is well tuned to discriminate water in both
rough and smooth regions of the ice sheet (Rippin, 2013; Jordan et al., 2017). Finally, Table 1 also
indicates that, in exceptional circumstances, $\Delta[R]$ > 12 dB could be generated in frozen/dry regions
that partially contain sandstone or till that is close to matching the permittivity of ice. However, if
present, these regions are likely to have indistinct bed-echoes and will not be included in the effective
coverage in Fig. 1(b).

## 2.6 Basal water distribution and robustness to attenuation model bias

The initial basal water predictions ($\sigma_{[R]}$ > 6 dB) are shown in Fig. 6 (red, blue and green data),
and correspond to $\sim$ 3.5 % of bins containing predominantly 'good' quality bed-echoes (Fig. 1(b)).
A full geographic analysis of the spatial distribution is performed in Sect. 3. To demonstrate the
robustness of the predictions, we performed a sensitivity analysis with respect to the modelled at-
tenuation correction $< N >$ (Sect. ) The analysis considered a series of increasingly large (uniform,
multiplicative) perturbations to $< N >$ and then tested if $\sigma_{[R]}$ > 6 dB also held for the perturbed
model. Examples of 'persistent' water predictions for $\pm$ 20 % (red and green data) and $\pm$ 50 % (red
data) perturbations to $< N >$ are indicated. As the perturbation size increases this results in a slight
decrease in the overall percentage of water predictions (corresponding to $\sim$ 2.6 % and 2.1 % of the
along-track bins for $\pm$ 20 % and $\pm$ 50 % respectively).

The sensitivity analysis tests the robustness of the water predictions to a number of different phys-
ical scenarios. Firstly, inherent bias in the Arrhenius equation parameters. For example, an empirical
correction similar to the uniform perturbation considered in Fig. 6 was proposed by MacGregor et al.
(2015b) to model unaccounted frequency-dependence in the electrical conductivity. Secondly, bias
in the model temperature field ($< N >$ is approximately equivalent to depth-averaged temperature).
Thirdly, bias due to assumed ionic concentration values. It is hard to formally quantify the possible
range of these uncertainties but, based upon solution variability for $< N >$ using ice-sheet model
temperature fields (Jordan et al., 2016), $\pm$ 20 % is a reasonable estimate for temperature related un-
certainty. Subsequently, in the comparison with other data sets in Sect. 3 the subset of red and green



points in Fig. 6 is used. Inherent bias in the Arrhenius equation parameters could be significantly
higher than temperature uncertainty (MacGregor et al., 2015b). However, since the spatial structure
for the basal water distribution under the $\pm$ 50 % perturbation is largely preserved, this is unlikely
to significantly alter the conclusions that are drawn.

It is important to emphasise the robustness of $\sigma_{[R]}$ with respect to uncertainty/model bias in
$<N>$ (particularly compared with bed-echo reflectivity, $[R]$). An analogous sensitivity analysis
by Jordan et al. (2016) demonstrated that systematic over and underestimates in $<N>$ lead to pro-
nounced ice thickness-correlated biases in the distribution for $[R]$ in northern Greenland (Fig. B1 in
the original paper).

## 3 Results

The basal water distribution is now compared with existing analyses for the basal thermal state
(MacGregor et al., 2016) (Sect. 3.1), geothermal heat flux (GHF) (Shapiro and Ritzwoller, 2004;
Fox Maule et al., 2009; Martos et al., In revision) (Sect. 3.2), bed topography and subglacial flow
paths (Morlighem et al., 2017) (Sect. 3.3), and ice surface speed (Joughin et al., 2010, 2016) (Sect.
3.4). The basal water predictions are always indicated by red circles. In regional zoom plots the
circles are fixed to be 5 km in diameter (a true representation of the along-track window size and
the effective resolution of the radar method). In ice-sheet scale plots the buffer size of the water
predictions are increased for visualisation purposes. The radar flight-tracks represent where there
are 'good' bed-echoes (Fig. 1(b)), and hence indicate the effective coverage.

In interpreting the maps it is important to emphasise that the basal water predictions correspond
to a subset of flight-track data where basal water is present (specifically, where there are rapid transi-
tions in bed material properties). They subsequently act as a constraint upon the distribution of basal
water rather than being a fully comprehensive flight-track map for water extent. Additionally, since
the vast majority of the radar measurements were collected before the onset of summer surface melt,
to a first approximation, the basal water predictions correspond to the winter storage configuration.

### 3.1 Comparison between basal water distribution and basal thermal state synthesis

In Fig. 7(a) the basal water predictions are underlain by the basal thermal state synthesis (frozen/thawed
likelihood) map by MacGregor et al. (2016). The synthesis employed four independent methods: (i)
assessment of thermomechanical model temperature fields, (ii) basal melting inferred from radios-
tratigraphy, (iii) basal motion inferred from surface velocity, (iv) basal motion inferred from surface
texture. The four methods were then equally weighted, leading to a likelihood map for frozen beds,
thawed beds, and uncertain regions. Importantly, the prediction did not utilise radar bed-echo data
and is therefore independent of our basal water predictions.





The reflectivity variability water diagnostic enables a positive discrimination of basal thaw, since $\sigma_{[R]} > 6$ dB is deemed as a sufficient (but not necessary) criteria for basal water. Positive discrimi-
nation of frozen regions is not, however, possible. This is because low reflectivity variability ($\sigma_{[R]} <$ 6 dB) could correspond to many different scenarios: a frozen region, a drier region at or above the PMP, or a wet region that is smoothly varying in bed-echo reflectivity. Since basal water enables a positive discrimination of thaw, red circles in likely thawed (pink) regions indicate agreement and red circles in likely frozen (blue) regions indicate disagreement with the basal thermal state synthe-
sis. Absence of basal water in likely frozen regions is an indicator of general consistency between the two methods.

There is general agreement (water in predicted thawed regions) for the beds of major outlet glaciers and their upstream regions. This includes Helheim, Kangerlussuaq, Jakobshavn, and the other fast flowing regions identified in Fig. 1(c). There is also general agreement between basal
water and the extent of predicted thaw in the NEGIS drainage basin. Major regions of disagreement (water in predicted frozen regions) are highlighted in the zoom plots, Fig. 7(b)-(f). The most obvious disagreement is the quasi-linear 'corridor' of basal water in the north-central ice sheet (Fig. 7(d)). This feature tracks close to the central ice divides and extends from the NorthGRIP region in the south toward Petermann glacier in the north. There are also noticeable areas of disagreement to the
north and east of the Camp Century borehole (Fig. 7(b)), in the far north (Fig. 7(c)), to the east of GRIP (Fig. 7(e)), and around Kangerlussuaq (Fig. 7(f)). There is also an absence of water in many predicted frozen regions indicating consistency. This includes parts of the southern interior, north of the NEGIS drainage basin, and the majority of the interior region between the Camp Century and NEEM boreholes.

**3.2    Comparison between basal water distribution and geothermal heat flux models**

The basal temperature of glacier ice is governed by GHF, strain heating from internal deformation, frictional heating, and diffusive and advective heat transport (e.g. van der Veen (2013)). In the interior of the ice sheet, close to the ice divides, GHF and vertical diffusion are the dominant processes which influence basal temperature. In this scenario, the thermodynamic (temperature) equation can
be approximated by the classical Robin model which predicts that basal melting occurs when GHF is above a certain threshold. However, more comprehensively determining the minimum GHF forcing required to produce basal melt requires coupling to ice-sheet flow models, and is anticipated to be $\sim$ 55-70 mW m$^{-2}$ in the interior of the ice sheet (Dahl-Jensen et al., 2003; Greve, 2005; Buchardt and Dahl-Jensen, 2007). In the water-GHF comparison we therefore define 'elevated' GHF (i.e. likely to
produce basal melt) as $> 60$ mW m$^{-2}$. This definition is also informed by the lower range of values, (37-50 mW m$^{-2}$) that are typically associated with non-altered ancient continental crust (Artemieva, 2006; Rogozhina et al., 2016).



In Fig. 8 the basal water predictions are underlain by three different GHF models: the seismic model by Shapiro and Ritzwoller (2004), and two models derived from magnetic anomalies by Fox Maule et al. (2009); Martos et al. (In revision). The GHF model by Shapiro and Ritzwoller (2004) is based upon the correlation between a 3D tomographic model of the crust and mantle temperature. The GHF models by Fox Maule et al. (2009); Martos et al. (In revision) are based on a thermal model of the lithosphere with the lower boundary defined by the Curie depth which is determined from magnetic anomalies. Martos et al. (2017) further describes this approach and the additional spectral processing method used to produce Fig. 8(c). An older tectonic GHF model by (Pollack et al., 1993) is not considered and a spatial plot for the GrIS can be found in Rogozhina et al. (2012) along with a discussion of the caveats of the different types of model. A summary of local GHF estimates using borehole temperature profiles and thermomechanical model inversions (Weertman, 1968; Dahl-Jensen et al., 1998, 2003; Greve, 2005; Buchardt and Dahl-Jensen, 2007; Petrunin et al., 2013) are provided by Rezvanbehbahani et al. (2017); Martos et al. (In revision), and demonstrate general consistency between Fig. 8(c) and local estimates at GRIP, NEEM, NorthGRIP and Camp Century. Local estimates of GHF at Dye 3 ($\sim$ 20-25 mW m$^{-2}$) are significantly lower than all three GHF models.

In interpreting Fig. 8, we limit the comparison to the ice sheet interior where the spatial correlation between GHF and basal water should be strongest. The model by Shapiro and Ritzwoller (2004), Fig. 8(a), predicts low GHF ($<$ 60 mW m $^{-2}$) over the vast majority of the central and northern interior. There is therefore no correlation between elevated GHF and basal water. The model by Fox Maule et al. (2009), Fig. 8(b), predicts elevated GHF around GRIP and the southern and eastern boundaries of the NEGIS basin and basal water is also present in this region. There is, however, no correlation between elevated GHF and basal water along the ice divides north of NorthGRIP. The model by Martos et al. (In revision), Fig. 8(c), exhibits strong overall spatial correlation between basal water and elevated GHF in the interior of the northern ice sheet. Notably, there is a striking correlation between elevated GHF and the quasi-linear 'corridor' of basal water that extends from NorthGRIP toward Petermann glacier. All three models predict regions of elevated GHF in the southern interior including the Dye 3 region. However, there is only isolated radar evidence for basal water.

### 3.3 Comparison between basal water distribution, bed topography and subglacial flow paths

In Fig. 9 the basal water predictions are underlain by the most recent Greenland bed topography digital elevation model (DEM) (Morlighem et al., 2017). To motivate further discussion about water storage locations and hydrological connectivity, a predicted subglacial flow path network is also included. The network structure is governed by gradients in the hydraulic pressure potential (Shreve, 1972) which was calculated using the bed elevation and ice thickness surfaces at a grid cell resolution of 600 m (derived from Morlighem et al. (2017)). The flow-routing algorithm was implemented in ArcGIS using the inbuilt flow accumulation tool and the hydrological sink filling procedure (Jenson





and Domingue, 1988; Tarboton et al., 1991; Planchon and Darboux, 2002). Likely hydrological flow
paths were identified by excluding flow paths where fewer than 50 neighbouring cells cumulatively
contribute to a given location.

Fig. 9 demonstrates that the vast majority of the basal water predictions are well aligned with
predicted subglacial flow paths. This alignment is most visually pronounced toward the margins and
zoom plots are shown for the Petermann catchment in Fig. 9(b) and northwestern margins in Fig.
9(c). Fig. 9(b) also demonstrates that basal water is present along sections of the 'mega-canyon'
feature identified by Bamber et al. (2013b) - for example, north west of the intersection o (80°N,
50°W). In the interior of the ice sheet, where the horizontal gradients in ice thickness are small, local
minima/sinks in the hydraulic potential surface should correlate with topographic depressions. The
water storage locations in the interior generally conform to this behaviour (Fig. 9(d)).

**3.4    Comparison between basal water distribution and ice surface speed**

In Fig. 10 the basal water predictions are underlain by ice surface speed (Joughin et al., 2016) which
is based upon a temporal average from 1st December 1995 to 31 October 2015. The ice surface speed
is determined using interferometric synthetic aperture radar (InSAR) as described in Joughin et al.
(2010). Whilst there is a complex overall relationship between basal water and ice velocity, there are
some clear spatial patterns. Notably, in the topographically less constrained northern and western
outlet glaciers, basal water is often concentrated in the fast-flow onset regions and tributaries whilst
it is absent from the main trunks. This behaviour is particularly evident for the Petermann glacier
catchment (Fig. 10(b)). In the topographically more constrained southeastern outlet glaciers, there
is widespread evidence for basal water storage in both the fast flowing glacial troughs and upstream
regions. This includes both the Kangerlussuaq catchment and the tight network of subglacial troughs
to the south (Fig. 10(c)), and the Helheim catchment (Fig. 10(d)).

In the interior of the ice sheet basal water is predicted near to the head of the NEGIS ice stream.
However, basal water is also predicted in some of the slowest flowing regions of the ice sheet interior.
Notably, close to the central ice divides between NorthGRIP and Petermann and north east of GRIP.

**4    Discussion**

**4.1    Basal water, basal thermal state and temperature**

The basal water distribution in this study and the basal thermal state synthesis by MacGregor et al.
(2016) represent two independent approaches to predict where the bed beneath the GrIS is thawed.
There is greatest agreement (basal water in likely thawed regions identified by MacGregor et al.
(2016)) toward the ice margins where ice surface speed is generally higher. The most noticeable
regions of disagreement (basal water in likely frozen regions identified by MacGregor et al. (2016))
all occur where the ice surface speed is low. This includes the north-central ice-divide (Fig. 7(d)),





the region east of GRIP (Fig. 7(e)), and the region west of Kangerlussuaq (Fig. 7(f)). The regions
of agreement/disagreement are, perhaps, unsurprising, since three of the four methods employed by

MacGregor et al. (2016) - ice-velocity, surface texture and radiostratigraphy - associate basal thaw
with present (or past) ice sheet motion. In general, a thawed bed is a necessary (but not sufficient)
condition for appreciable basal motion, and there is likely to be a subset of thawed regions where
basal motion is negligible. This subset naturally incorporates water/thaw near the ice divides (since
driving stress is low), and in the eastern ice sheet (since ice flow is topographically constrained).

Another key difference between the water predictions in this study and the thaw predictions by
MacGregor et al. (2016) is that their study employed techniques better tuned to identify continu-
ous regions of basal thaw, whereas the basal water predictions are localised. This provides another
means to reconcile regions of disagreement, since in some instances the basal water predictions may
correspond to localised patches above the PMP in an otherwise frozen region. A final explanation for

discrepancies, is that the model temperature fields included in the basal thermal state synthesis were
often tuned around knowledge of GHF at the time (i.e. Shapiro and Ritzwoller (2004); Fox Maule
et al. (2009)).

There is no evidence for basal water at the location of the temperature boreholes, which, based
upon the resolution of our method, corresponds to within 5 km. Since high reflectivity variability is

not necessary for thaw this is consistent with both frozen and thawed borehole temperatures. Water
is, however, observed fairly close to two frozen boreholes: $\sim 10$ km south of GRIP and $\sim 7$ km
northeast of Camp Century (Fig. 7). At GRIP this is less surprising, since the basal temperature
is 6 degrees below the PMP (Dahl-Jensen et al., 1998; MacGregor et al., 2016) and GHF is likely
to be elevated in this region (Fig. 8). The basal water predictions near Camp Century are more

surprising, since in the late 1960s basal temperatures were measured to be 11.8 degrees below the
PMP (Weertman, 1968; MacGregor et al., 2016). One possible explanation, which was recently
invoked to explain the presence of a lake less than 10 km from South Pole (where the bed is frozen),
is that the basal water is yet to reach thermal equilibrium (Beem et al., 2017). Another possible
explanation is that the presence of hypersaline water could result in a depression of the PMP. This

situation arises at Lake Vida in East Antarctica (where liquid water exists at -13 °C) (Murray et al.,
2012) and at Devon Ice Cap in the Canadian Arctic (Rutishauser et al., 2017).

### 4.2 Basal water and geothermal heat flux

The comparison between basal water and the different GHF models in Fig. 8 (Shapiro and Ritz-
woller, 2004; Fox Maule et al., 2009; Martos et al., In revision) demonstrates greatest consistency

with the distribution by Martos et al. (In revision). Notably there is a pronounced spatial correlation
between elevated GHF and the new predictions of basal thaw in the northern ice sheet. A recent
machine learning derived map for GHF beneath Greenland by (Rezvanbehbahani et al., 2017) is
also consistent with there being extensive basal thaw in this region. However, establishing definitive



attribution of regions of the basal melt to GHF forcing (rather than frictional and strain heating, low
advection from colder ice above, and/or surface melt water storage) will require integration with
thermomechanical ice-sheet models. The basal water predictions could also be used as a constraint
in a wide variety of other numerical modelling contexts. Experiments with 3D models to reconstruct
the full ice temperature history over the last glacial cycle(s) can constrain the minimum GHF re-
quired to produce basal melting at the predicted basal water locations (Huybrechts, 1996). Other
studies include investigating the sensitivity of ice-sheet dynamics to the thermal boundary condi-
tion (Seroussi et al., 2013) or basal lubrication (Shannon et al., 2013), and thermal models of the
underlying lithosphere (Rogozhina et al., 2016).

Recent analyses imply that much of the spatial variation in GHF beneath the northern GrIS can be
explained by Greenland's passage over the Iceland mantle plume between roughly 35 and 80 million
years ago (Rogozhina et al., 2016; Martos et al., In revision). The magnetic GHF map in Fig. 8(c),
alongside gravity data (Bouger anomalies), was recently used to by Martos et al. (In revision), to
infer the most likely passage of the hotspot track. The most likely predicted path (corresponding to
going forwards in geological time) follows the quasi-linear region of elevated GHF in Fig. 8(c) from
Petermann glacier to NorthGRIP, and follows a path previously anticipated by Forsyth et al. (1986).
The spatial correlation between elevated GHF and the quasi-linear basal water 'corridor' provides
an additional source of evidence for the predicted path.

### 4.3  Basal water, bed topography and subglacial flow paths

There is growing evidence that much of the present day subglacial flow path network beneath the
GrIS is paleofluvial in origin. This includes the dendritic flow path networks in the Jakobshavn
(Cooper et al., 2016) and Humboldt catchments (Livingstone et al., 2017), along with the prominent
'mega-canyon' feature which extends from the NorthGRIP region in the south to Petermann glacier
in the north (Bamber et al., 2013b). The comparison between the predicted flow paths and basal water
in Fig. 9 enables a revised assessment of the hydrological flow paths that are likely to be utilised in
the contemporary ice sheet. For example, when the mega canyon was first identified by Bamber
et al. (2013b), accompanying flow routing analysis demonstrated that basal water originating in the
Petermann catchment is likely to route through sections of the canyon toward the ice-sheet margins.
Fig. 9(b) supports this hypothesis, since the there is evidence for basal water along the majority of the
canyon. However, it is important to stress that more rigorously assessing hydrological connectivity
will require incorporation of DEM uncertainty when performing the flow routing (e.g. Schroeder
et al. (2014)) and use of a coupled hydrological ice flow model (e.g. Le Brocq et al. (2009)).

### 4.4  Basal water and ice-sheet motion

Both observational (e.g. Moon et al. (2014); Tedstone et al. (2013)) and theoretical studies (e.g.
Creyts and Schoof (2009); Schoof (2010)) point toward a complex spatio-temporal relationship be-





tween basal water and ice surface speed in fast flowing regions of the ice sheet. This ultimately

depends upon the details of how the subglacial drainage system responds to surface melt water. In interpreting the relationship between basal water and ice surface speed in Fig. 10 it is therefore essential to re-emphasise that the basal water predictions correspond to the winter storage (pre surface melt) configuration. Nevertheless, the apparent absence of water in the main trunks of the fast moving outlet glaciers (e.g Petermann in Fig. 10(b)) can potentially be linked to our understanding

of the seasonal evolution of the subglacial drainage system. Specifically, in regions where there is significant surface melt-water forcing, efficient channelised drainage systems can form during the summer months, which can result in a substantial lack of winter-time water storage in faster moving subglacial troughs (Chu et al., 2016).

In addition to basal water and temperature, spatial variation in the underlying geology and lithol-

ogy of the GrIS (notably, presence or absence of deformable sediment) will also influence ice-sheet motion. It is widely anticipated that much of the interior of the ice sheet is underlain by hard pre-Cambrian rocks, with more limited sedimentary deposits toward the margins (Dawes, 2009; Henriksen, 2008), and in the NEGIS drainage basin (Christianson et al., 2014). It is therefore entirely plausible that much of the basal water predicted in the interior lies upon a hard undeformable bed

(particularly in the context of the igneous rock that would be associated with the geological remnants of the Iceland hotspot track) and therefore experiences little motion due to bed deformation.

### 4.5 Comparison with past RES analyses of basal water in Greenland

Despite acknowledged calibration issues, due to both variable radar system performance and spatial variation in attenuation, the bed-echo reflectivity analysis of 1990s PARCA RES data by Layberry

and Bamber (2001) anticipated some of the water predictions in this study. This includes prior basal water predictions in the NEGIS onset region, and the upstream regions of Kangerlussuaq, Petermann and Humboldt glaciers.

There is a mixed agreement between the basal water predictions in this study and Oswald and Gogineni (2008, 2012) who performed joint bed-echo reflectivity/scattering analysis of the 1990s

PARCA data. In general, better agreement with our results occurs in smoother topographic regions in the ice-sheet interior such as close to the NorthGRIP borehole. Since the effects of spatial bias due to attenuation uncertainty are lower in the interior of the ice sheet, this is where bed-echo reflectivity as a water diagnostic should be more robust. Additionally, the water detection method proposed by Oswald and Gogineni (2008, 2012) will generally not be able to discriminate water in many outlet

glaciers and tributaries including Petermann and the northwestern margins (Jordan et al., 2017). This is because these regions tend to exhibit a diffuse scattering signature (associated with fine-scale roughness) whereas the method proposed by Oswald and Gogineni (2008, 2012) is specifically tuned to detect water bodies that exhibit a spatially continuous, (near-) specular scattering signature.





In addition to bed-echo reflectivity, the 'freeze-on' features identified in radargrams also provide
evidence for the presence of basal water (Bell et al., 2014). In northern Greenland, these freeze-on
features are typically aligned with basal water predicted in fast flow initiation regions (e.g. Petermann
and the northern tributaries of NEGIS).

### 4.6  Limitations of bed-echo reflectivity variability as a RES technique to detect basal water

Bed-echo reflectivity variability provides a practical way to automate the detection of a subset of
basal water with high confidence at the ice-sheet scale (specifically, finite water bodies with sharp
horizontal gradients in water content). It is, however, important to note that the approach will fail
to identify basal water/wet regions with a homogeneous dielectric and reflective character. This in-
cludes the centre of large subglacial lakes (based upon the resolution of our method lakes greater
than 5 km in horizontal extent) and regions of more uniformly saturated subglacial till. Since all
identified subglacial lakes in Greenland are < 5 km in horizontal extent (Palmer et al., 2013; Howat
et al., 2015; Palmer et al., 2015; Willis et al., 2015) we believe that the former scenario is likely
to be rare. However, extensive regions of saturated till that evade detection are likely to be present,
particularly beneath larger outlet glaciers such as Petermann. If focusing on these regions (or other
glaciogically similar regions of Antarctica) a suite of existing RES techniques to detect and char-
acterise basal water (e.g. Peters et al. (2005); Jacobel et al. (2009); Schroeder et al. (2013); Young
et al. (2016)) are better suited.

### 5  Summary and conclusions

This study placed a spatially comprehensive observational constraint upon the basal water distribu-
tion beneath the GrIS and hence regions of the bed at or above the PMP of ice. The distribution of
basal water is influenced by, and has influence upon, multiple ice-sheet and subglacial properties and
processes. Subsequently, with a focus upon ice-sheet scale behaviour, we performed an exploratory
comparison with related data sets for the GrIS. This included an up-to-date synthesis for the basal
thermal state (MacGregor et al., 2016), three different GHF model distributions (Shapiro and Ritz-
woller, 2004; Fox Maule et al., 2009; Martos et al., In revision), bed topography (Morlighem et al.,
2017) and predicted subglacial flow paths, and ice surface speed (Joughin et al., 2010, 2016).

Central to the methods in the study was the use of bed-echo reflectivity variability (rather than
bed-echo reflectivity) as a RES diagnostic for basal water. Our use of this diagnostic was motivated
by its insensitivity to radar attenuation at the ice-sheet scale, and the pragmatic advantages when
performing data combination for multiple RES field campaigns. The reflectivity variability diagnos-
tic is, however, only able to detect wet to dry (or wet to frozen) transitions in bed material. It will
therefore need to be combined with other information to fully map basal water extent and classify
basal water bodies.





There was much agreement between the basal water distribution and the thawed marginal regions predicted by MacGregor et al. (2016). However, we did identify regions of basal water/thaw in the interior of the ice sheet that were previously classed as likely to be frozen. The most extensive 'new' region of predicted thaw is a quasi-linear 'corridor' feature which extends from NorthGRIP in the south to Petermann in the north. This feature, and the majority of basal water in the northern interior, spatially correlate with elevated GHF inferred from magnetic data by Martos et al. (In revision).

The comparison with bed topography (Morlighem et al., 2017) and predicted flow paths, demonstrated good overall agreement between the basal water storage locations and the geometric constraints imposed by the hydrological pressure potential. However, many of the basal water predictions in the ice-sheet interior occur where ice surface speed (and hence basal motion) is negligible. One plausible explanation is that much of the interior lies upon a hard and undeformable bed. Future investigation of basal control upon GrIS dynamics, should integrate information about basal water and the basal thermal state with better constraints upon bed lithology and geology.





**Data availability**

The RES data are available from CReSIS at https://data.cresis.ku.edu/data/ rds/ and are documented in Paden (2015). The profile in Fig. 2(a) is from data segment 2012050804 from the 2012 P3 season and the profile in Fig. 2(b) is from data segment 2014051601 from the 2014 P3 season. The Green-
land basal thermal state synthesis (MacGregor et al., 2016), ice thickness and topography data sets (BedMachine V3) (Morlighem et al., 2017), and ice surface speed (Joughin et al., 2016), are archived by NSIDC at http://dx.doi.org/10.5067/ R4MWDWWUWQF9, https://nsidc.org/data/idbmg4 and https://nsidc.org/data/NSIDC-0670/versions/1 respectively. The GHF maps by Fox Maule et al. (2009) and Shapiro and Ritzwoller (2004) and available at http://www.dmi.dk/dmi/dkc09-09.pdf and
http://ciei.colorado/edu/∼nshapiro/MODEL/ASC_VERSION/index.html respectively.

Pending review, the basal water distribution will be provided as csv files on both a season-by-season basis and for the full (13 season) data set. The data columns will correspond to: (A) latitude, (B) longitude, (C) water binary value, at a 1 km along-track posting. The water binary value will correspond to: $1 \equiv$ a 5 km bin with water ($\sigma_{[R]} > 6$ dB, robust to a $\pm$ 20 % perturbation in $< N >$),
$0 \equiv$ a 5 km bin with a majority of good quality bed-echoes but no water, NaN $\equiv$ a 5 km bin with a majority of poor quality bed-echoes (interpreted as no coverage). A geotiff overlay for the water predictions and radar flight-tracks will also be provided.

**Author contributions**

The study was initiated by JLB and co-advised by MJS as part of the Basal Properties of Greenland
Project. TMJ wrote the paper, analysed the radar data, and, with advice from DMS, developed the water detection method. CNW and MAC developed the GIS mapping environment and modelled the subglacial flow paths. JDP and DMS both advised on radar data processing. YMM contributed the new heat flux distribution and advised on the interpretation of the heat flux models. PH contributed the GISM temperature field and advised on ice-sheet thermodynamics. All authors commented on
the paper draft and contributed to the interpretation of the science.

**Competing interests**

JLB is an advisory editor of The Cryosphere. The authors declare that they have no conflict of interests.

**Acknowledgments**

The project was primarily supported by UK NERC grant NE/M000869/1 as part of the Basal Prop-erties of Greenland project. T.M.J. was also supported by an EU Horizons 2020 grant 747336-





BRISRES-H2020-MSCA-IF-2016. M.A.C. was supported by the UK NERC grant NE/L002434/1
as part of the NERC Great Western Four + (GW4+) Doctoral Training Partnership. D.M.S. was sup-
ported by a grant from the NASA Cryospheric Sciences Program. We acknowledge the use of data
660   products from CReSIS generated with support from NASA grant NNX16AH54G. We would also
like to thank Winnie Chu, Stanford University, for her helpful discussions.



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




**Table 1.** Dielectric and reflective properties of subglacial materials based upon a compilation of past values by Bogorodsky et al. (1983); Martinez et al. (2001); Peters et al. (2005). The bulk values take into account typical ranges of saturation and porosity for the dielectric mixing of water and ice with the background material. The relative dielectric permittivity of ice is 3.15, which means that 'dry' (just the background dielectric) or 'frozen' (a mixture of the background dielectric with ice) produce a similar range for $[R]$.

| Bed material | Relative dielectric permittivity, $\epsilon$ | Reflectivity, $[R]$ (dB) |
|---|---|---|
| Ground water | 80 | -2 |
| Wet till | 10 to 30 | -11 to -6 |
| Wet sandstone | 5 to 10 | -19 to -11 |
| Dry/frozen granite | 5 | -19 |
| Dry/frozen limestone | 4 to 7 | -26 to -14 |
| Dry/frozen till | 2 to 6 | negligible to -19 |
| Dry/frozen sandstone | 2 to 3 | -37 to -16 |





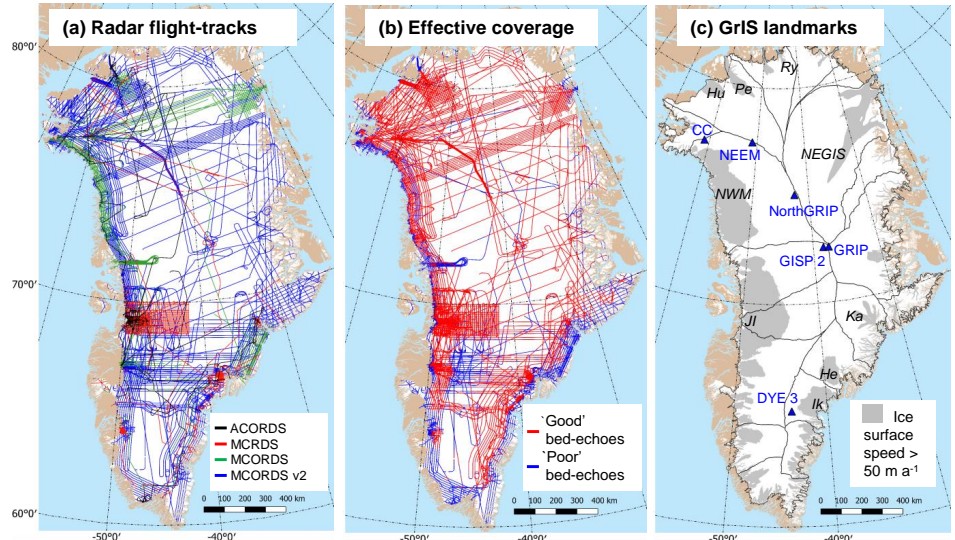

**Figure 1.** (a) Ice-penetrating radar flight-tracks for different CReSIS radar systems. (b) Effective coverage for 'good quality' radar bed-echoes (corresponding to peak power 10 dB above noise floor). (c) Summary of key GrIS landmarks: temperature boreholes, major drainage basin boundaries (Zwally et al., 2012), and major regions of fast flow identified from ice surface speed (Joughin et al., 2010, 2016). Abbreviations in (c) correspond to: Camp Century (*CC*), Humboldt (*Hu*), Petermann (*Pe*), Ryder (*Ry*), North East Greenland Ice Stream (*NEGIS*), northwestern margins (*NWM*), Jakobshavn Isbrae (*JI*), Kangerlussuaq (*Ka*), Helheim (*He*) and Ikertivaq (*Ik*). The projection is a Polar Stereographic North (70°N, 45°W) and is used in all future plots.



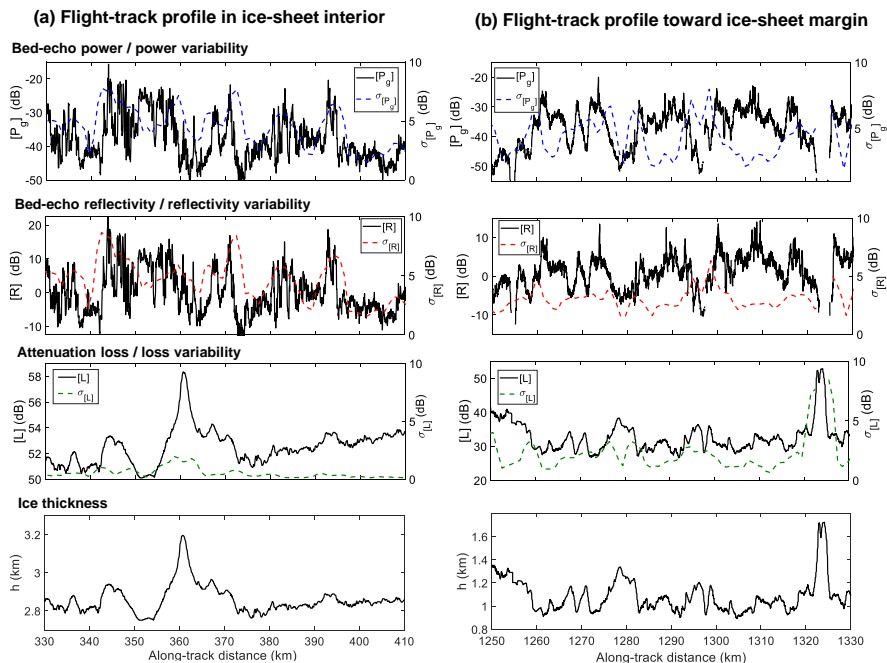

**Figure 2.** Example flight-track profiles for: bed-echo power and variability, $[P_g]$ and $\sigma_{[P_g]}$, bed-echo reflectivity and variability, $[R]$ and $\sigma_{[R]}$, attenuation loss and variability, $[L]$, $\sigma_{[L]}$, ice thickness, $h$. Example (a), left of figure, is from the north-central interior of the ice sheet and (b), right of figure, is from the northwestern margins (locations both shown in Fig. 3). The values for $[R]$ are relative and have a zero mean. The variability measures are all calculated at a 5 km length scale with 1 km posting.



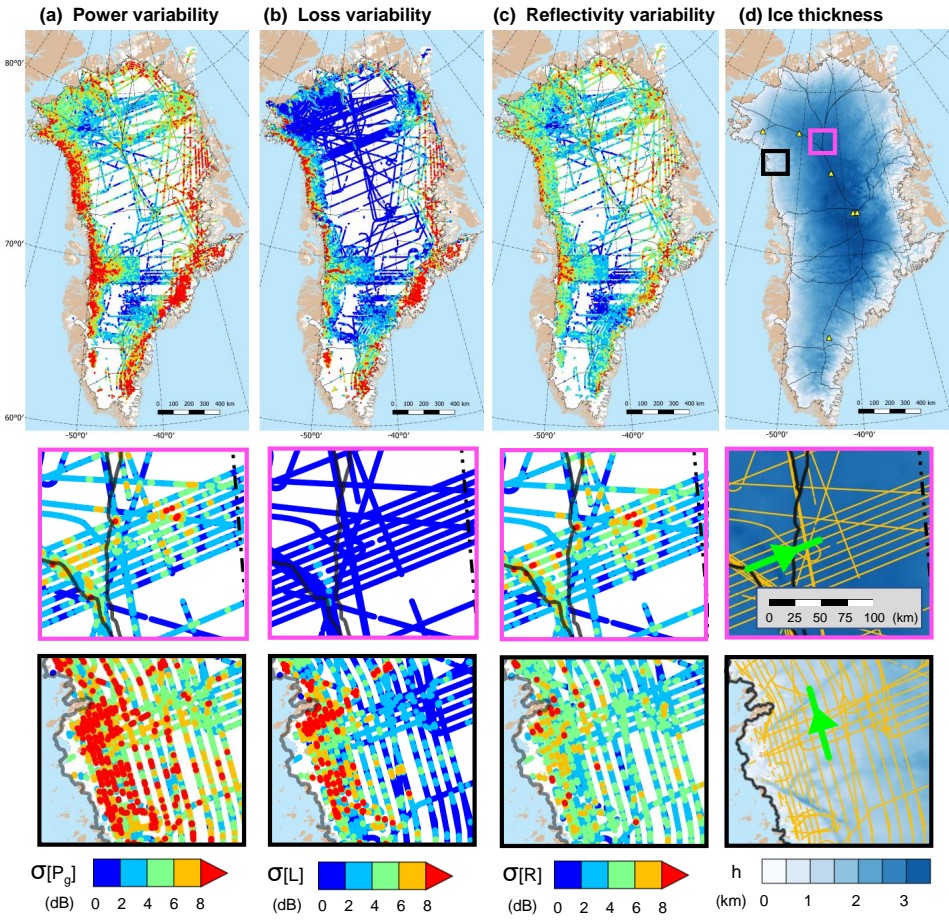

**Figure 3.** Spatial distributions for: (a) bed-echo power variability $\sigma_{[P_g]}$, (b) attenuation loss variability $\sigma_{[L]}$, (c) bed-echo reflectivity variability $\sigma_{[R]}$, (d) ice thickness, $h$. Zoom plots with flight-track data at true buffer size (5 km) are shown for the north-central ice sheet (pink bounding box, containing profile in Fig. 2(a)) and northwestern margins (black bounding box, containing profile in Fig. 2(b)). The profiles are indicated in bold green in the ice thickness zoom plots. In plots (a)-(c) higher variability data is stacked on top of lower variability data, which acts to emphasise higher variability. The zoom plots are all have the same scale ($\times 8$ the resolution of the ice-sheet scale plots).





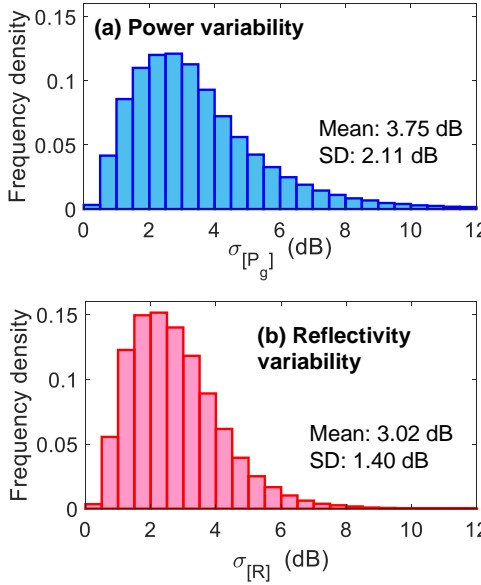

**Figure 4.** Frequency distributions for: (a) bed-echo power variability, $\sigma_{[P_g]}$ (corresponding to Fig. 3(a)); (b) bed-echo reflectivity variability, $\sigma_{[R]}$ (corresponding to Fig. 3(c)). Later in the study $\sigma_{[R]} > 6$ dB used as threshold criteria for diagnosing the presence of basal water.

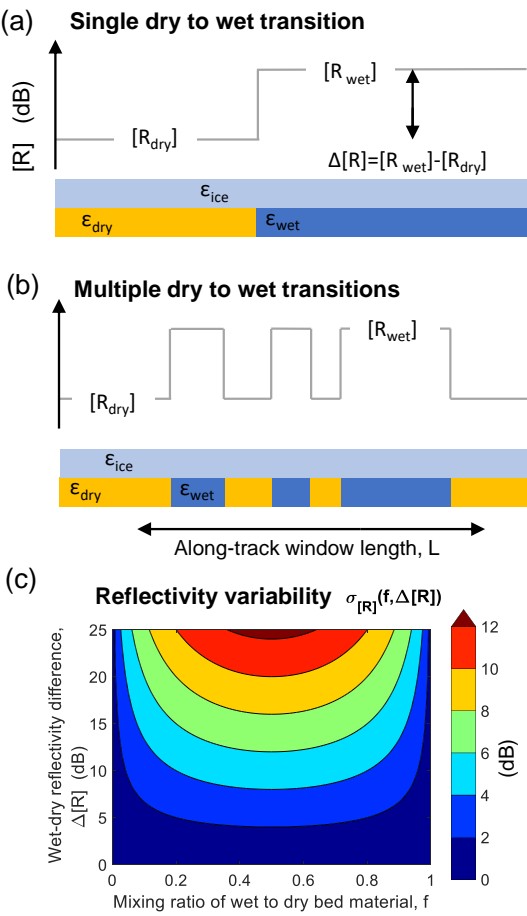

**Figure 5.** Interpretation of bed-echo reflectivity variability, $\sigma_{[R]}$, as a diagnostic for basal water. (a,b) Schematics of the two-state dielectric model for single and multiple along-track transitions in dry to wet bed material. Both scenarios are identically parameterised by the wet-dry mixing ratio $f$ (visually, the fraction of blue to yellow) and wet-dry reflectivity difference, $\Delta[R] = [R_{wet}] - [R_{dry}]$. (c) Phase-space plot for $\sigma_{[R]}$ as function of $f$ and $\Delta[R]$. $\sigma_{[R]} > 6$ dB is used as a threshold for positive discrimination of basal water (corresponding to green, red and yellow regions).



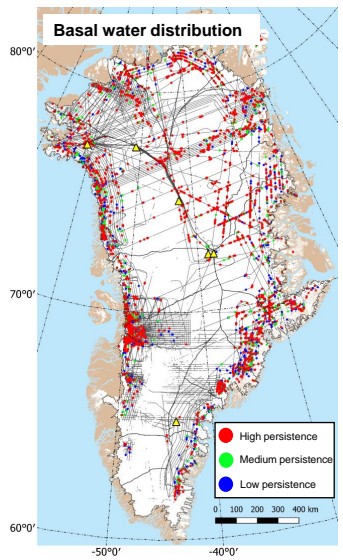

**Figure 6.** Basal water distribution and robustness to perturbations in the attenuation rate estimate, $< N >$. The original predictions ($\sigma_{[R]} > 6$ dB) are represented by all three colours. 'Persistent' water predictions ($\sigma_{[R]} > 6$ dB for $\pm 20$ % and $\pm 50$ % perturbations to $< N >$) are indicated by the subset of green and red points, and the subset of red points respectively. The subset of red and green points is used in the rest of the paper.



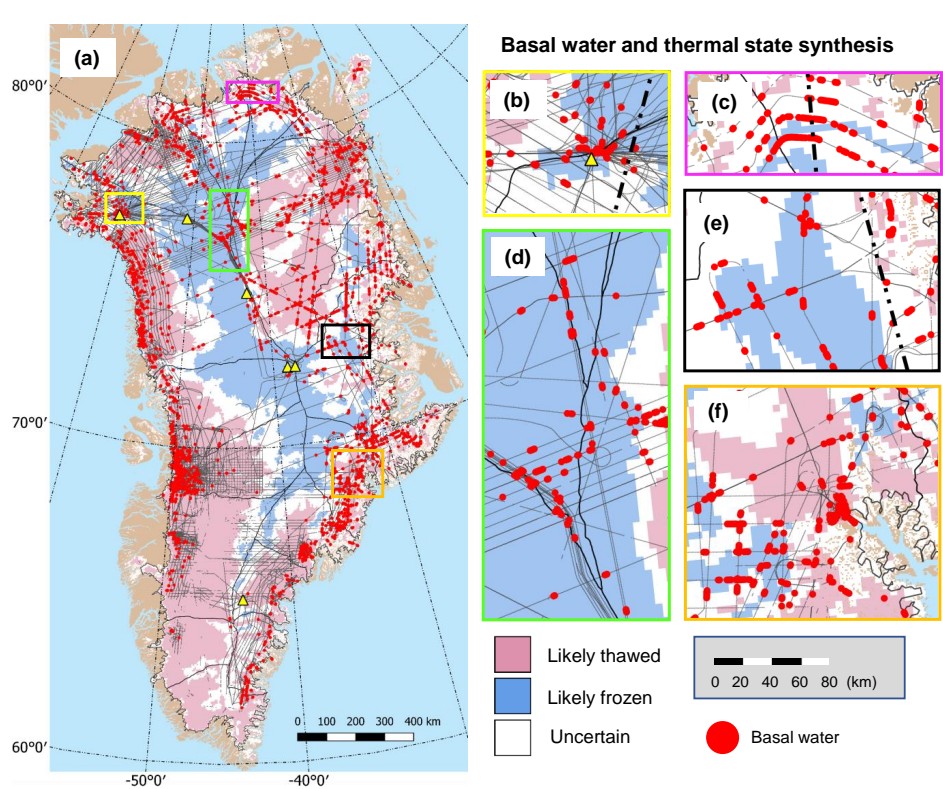

**Figure 7.** Comparison between basal water distribution and basal thermal state synthesis by MacGregor et al. (2016). (a) Ice-sheet scale. Major regions of disagreement (water in likely frozen regions) are highlighted in the zoom plots. (b) Camp Century. (c) Far north. (d) North-central ice sheet. (e) East of GRIP. (f) Around Kangerlussuaq. The zoom plots all have the same scale ($\times$ 5 the resolution of (a)).





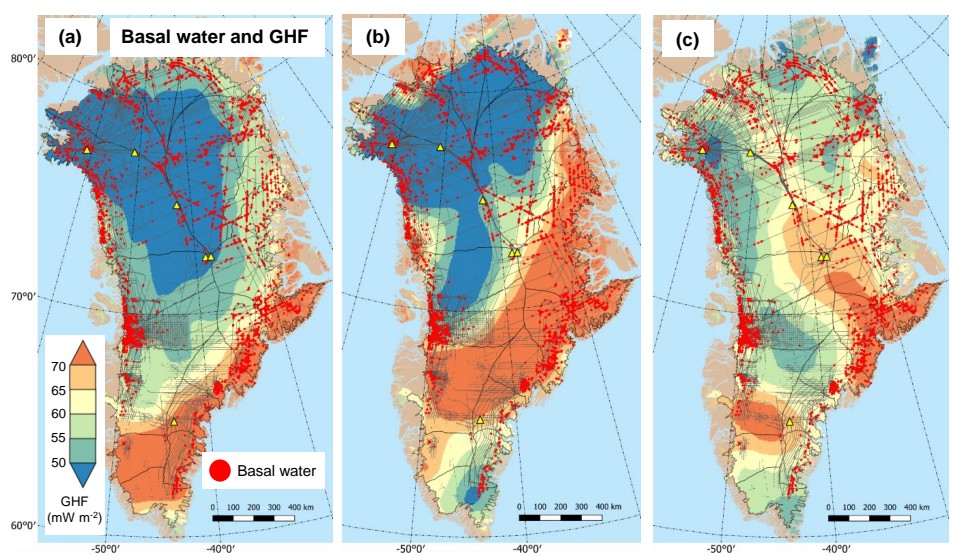

**Figure 8.** Comparison between basal water distribution and geothermal heat flux (GHF) models. (a) Seismic GHF model by Shapiro and Ritzwoller (2004). (b) Magnetic GHF model by Fox Maule et al. (2009) using satellite data. (c) Magnetic GHF model by Martos et al. (In revision) derived from spectral methods using airborne data. The colour bar scale is the same in all plots and is truncated to emphasise the spatial variation in plot (c).





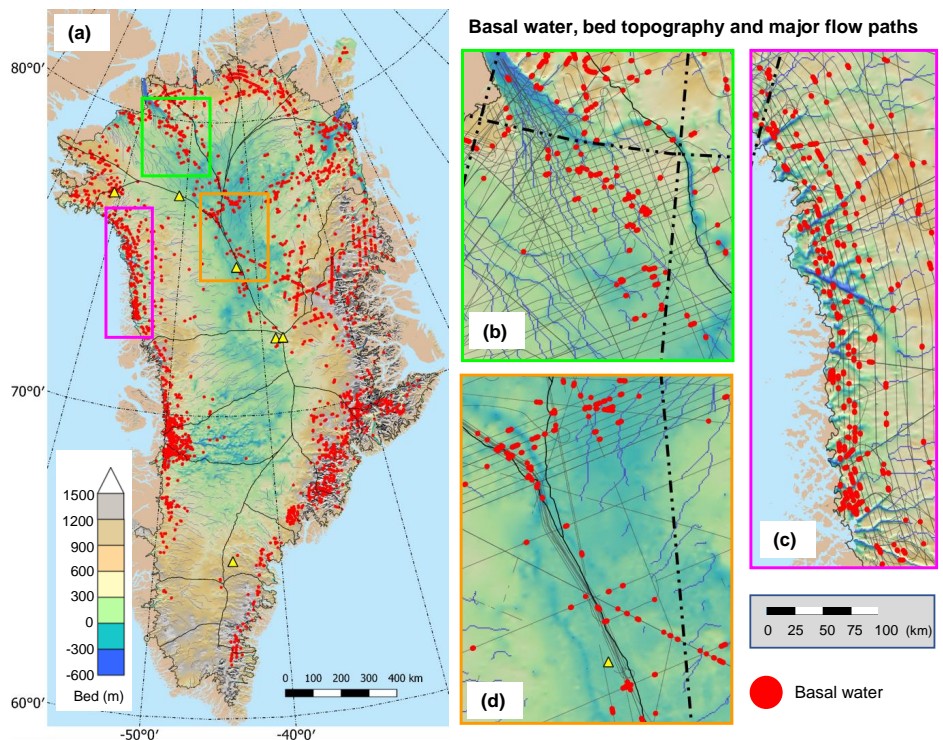

**Figure 9.** Comparison between basal water distribution, bed topography (Morlighem et al., 2017), and major subglacial flow paths (blue lines). (a) Ice-sheet scale. (b) Petermann catchment. (c) Northwestern margins. (d) North-central ice sheet. To improve clarity, the radar flight-tracks are removed from (a) and a hillshade is applied to the bed topography. The zoom plots are all have the same scale (× 4 the resolution of (a)).



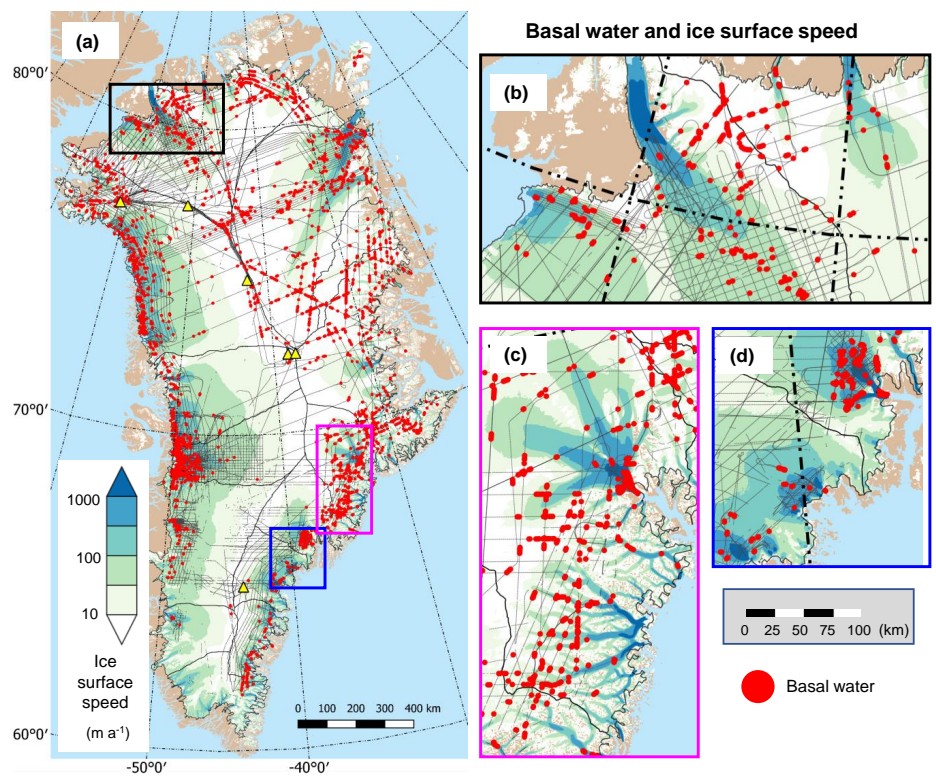

**Figure 10.** Comparison between basal water distribution and ice surface speed (Joughin et al., 2010, 2016) (logarithmic-scale). (a) Ice-sheet scale. (b) Humboldt, Petermann and Ryder. (c) Kangerlussuaq and region to south. (d) Helheim (north of plot) and Ikertivaq (southwest of plot). The zoom plots are all have the same scale (× 4 the resolution of (a).)