# Peer review of "A constraint upon the basal water distribution and thermal state of the Greenland Ice Sheet from radar bed-echoes"

_The Cryosphere, 2018_

## Referee Comment (RC1) · RD Drews (Referee) · 17 May 2018

**Review of Jordan et al.: "A constraint upon basal water distribution and basal thermal state of the Greenland Ice Sheet from radar bed-echoes"**

**General Impression:**

Jordan et al. analyze an extensive radar dataset covering large parts of Greenland. They introduce a novel metric ("the bed reflectivity variability") to locate basal water at the ice-bed interface. This is potentially a sufficient (but not a necessary) condition for water at the ice-sheet base. The results can, for example, be used as a constraint for modelling the evolution of the Greenland Ice Sheet, which critically depends on the conditions at the basal boundary (wet vs. dry).

I have read this paper with great interest, and the authors do an excellent job in guiding the reader through the manuscript. Basic methodology is explained succinctly (incl. appropriate references), and novel parts are correspondingly highlighted and more detailed. All Figures are informative and of high quality. I am impressed with the scope of the analysis, which includes a very rich radar dataset, and the careful interpretation of the "basal reflectivity variability" as a new (but not the only) metric for basal water. In the following, I have a number of comments which should be addressed, and which hopefully will make the paper stronger. I believe that this paper will be useful for other researchers in the future, and apologize to the authors, and to the editor, for the delayed submission of this review.

Kind regards,

Reinhard Drews, Geosciences, University of Tübingen, Germany

**Comments:**

1. **Interpretation of basal reflectivity variability:**
   The authors make it clear (in l. 368) that reflectivity variability above the chosen threshold is a sufficient criterion for basal water. However, the converse argument (that is: low reflectivity variability, hence, absence of basal water) does not hold. It is very important that people using the results of this study are aware of this, and it should be mentioned more clearly (i.e. in the abstract and elsewhere). The importance of this point is highlighted because the authors themselves misinterpret their own results in this regard. The inference that "…basal water is often concentrated in the fast-flow onset regions and tributaries whilst it is ***absent*** from the main trunk." (l. 457) is not valid, because the absence of elevated bed reflectivity does not necessitate the absence of basal water (it could simply be homogenously distributed and thus not be visible with this metric). I see much potential for misinterpretation here, and the authors should describe the limitations of this novel metric in a more pronounced way.

2. **Derivation of basal reflectivity variability:**
   Does it matter that only log-transformed variables are used? I wonder this, because:

$$VAR(10\,log10(X)) \neq 10log10\big(VAR(X)\big)$$

I can see the convenience of the log-transform when interpreting the reflection amplitudes, but I am unsure if this causes problems when calculating variances (and variance of multiplicative variables). Is there an underlying assumption about the statistics/uncertainties that was not explicitly mentioned? I understand this is a somewhat diffuse comment.

3. **Crossing-Over Analysis of the bed reflectivity variability**
   The authors should do a crossing-over analysis of the bed reflectivity variability. This would strengthen the argument that birefringence and radar system specifics are small. It will also more clearly demonstrate the robustness of the new metric and highlight its advantages (which is that radar data collected over multiple field seasons and with various radar systems can efficiently be combined).

4. **Discrimination between water patches and smooth--rough transitions**
   Variability in bed roughness is a competing mechanism which would also result in elevated bed-reflectivity variability. The authors are aware of this and suggest that the threshold is well tuned to discriminate between these two scenarios. However, I did not fully understand why this is the case and the manuscript is in this regard unfortunately too vague (e.g. "..we demonstrate later …" (l. 309) but where is this actually done?). This is important, because interpretation of the "bed reflectivity variability" as a proxy for basal water is a central conclusion of this paper, and other options must be convincingly excluded.

5. **Temperature profile near ice divides**
   I disagree that in the interior of ice sheets GHF and vertical diffusion are the dominant processes. What about vertical advection? In Greenland the surface mass balance is significant in the interior and thus I would expect a (strongly) non-linear temperature profile with depth (which would not be the case if only diffusion was important). A quick-look at the NGRIP profile does confirm this. Can you comment on this?

6. **Attenuation correction using modelled temperatures**
   Briefly explain what the modelled temperature field of Goelzer et al., 2013 is based on, (how it compares to in-situ measurements), and how it may impact your results. Is it possible that temperature variations near the coast (where ice streaming is significant) are smoothed out, and thus do not correctly cancel the attenuation in your approach?

7. **Ice Fabric Variations**
   I agree that fabric variations are a small component in the overall backscattered power budget. However, could it be that the "corridor" along the central ice divide is to some degree linked to ice anisotropy? Across a divide, ice fabric can change abruptly (compared to your 5 km window) (e.g. Martin et al., J. Geophys. Res., 2009; Drews et al., J. Glac., 2012) and potentially this may impact the inferred "bed reflectivity variability".

**Minor Suggestions**

Maybe "A sufficient constraint upon basal water distribution beneath the Greenland Ice Sheet from radar bed echo variability" would be a better (and slightly shorter) title?

l. 320 There is a missing section number in the internal reference.

l. 580 Basal 'freeze-on' is one, but not the only explanation for the disturbances seen at larger depth. Concerns about this (e.g. Dow et al., Geophys. Res. Lett., 2018) or other explanations (e.g. Bons et al., Nat. Comms., 2016) should also be mentioned here.

Fig. 5b add y-label ([R] (db))

---

## Referee Comment (RC2) · Anonymous Referee #2 · 28 May 2018

Summary: This paper uses a RES diagnostic the author's term "bed-echo reflectivity variability" for the long archive of RES observations over Greenland to get at the distribution of basal water. They then proceed to compare this to various prediction for the distribution of subglacial water. It is comprehensive and thorough, however I cant help thinking its being presented as a lot more sophisticated than it actually is. High passing radar data has been a (justifiable) refuge of radioglaciologists since the C-130 TUD days, and thats basically what seems to be happening here - just in slow time rather than fast time.

Major issues: Novelty: Bed echo variability has been long used for characterizing the

basal interface (Neal, 1982; Peters et al., 2005; Carter et al., 2007), and the surface interface (Grima et al., 2014) - most of this literature is not mentioned from this context. For the most part, bed echo peak power variability has been used to indicate interface roughness. The authors here extend to very long length scales, and integrate in fast time over the echo to suppress roughness effects in an attempt to essentially map out subglacial water using an assumption of bimodal wet/dry distribution, to get around variability in attenuation that will inevitably bias absolute values.

Edge detection: The approach I feel is misnamed. From Figure 2 it seems clear that large scale changes dominate their analysis, and thus basically what the authors have is an edge detector. What they are finding is not so much variability as gradients. In order to get at the small scale variability indicated in figure 5b, they would have to high pass the data, which they are not doing. A multi-scale approach may be more productive to get at mixed media cases.

Statistics in dB space: I am concerned at the application of statistics in dB space, and think this needs to be better motivated. Due to the compression of the distribution of the echoes using the attenuation model, and the highly bimodal reflectivity of the bed, they 'get away with it' somewhat; however, I attach an jupyter notebook that attempts to illustrate the complexities of doing the variability statistics in dB using a synthetic fractal distribution (again, the hypothesized distribution will be more bimodal, but there will be a sensitivity long wave length errors in attenuation) and a bimodal distribution.

Calculation of sigma: It wasn't clear if you are taking the deviation in power of points separated by 5 km, or just taking the standard deviation of all points within the 5 km window.

Ice surface transmission losses: Surface losses due to roughness or near surface englacial water are not considered. I think for this paper they could be important, as they do not correlate with the predictions for whole ice sheet attenuation, and have the potential for sharp gradients. Surface and near surface losses should be addressed,
maybe just by a demonstration that they are negligible.

Minor issues: Data traceability: The authors need to emphases that these data are from the CSARP processor found on the KU website, and NOT the MDVR processed data found on the NSIDC website. The latter should not be used for quantitative analysis of the bed echo. NSIDC says this in their website, but a disclaimer here might help head off confusion.

Along track processing: There is very little detail on how azimuth processing has changed over time, and how this could effect the results. Also not clear: did the author do along track incoherent averaging as they did for earlier papers?

Power determination approach: The method for extracting aggregate power, and rejecting bad echoes, appears to have changed between Jordan et al 2016, 2017. In those works, a symmetrical window about the peak is taken, while in this work, a 10 dB threshold down from the peak power is used. A 10 dB SNR is used here, while in earlier work it appears to be a 17 dB threshold is used. The change needs to be better justified, and the opportunity taken to explain if there is any impact of the results of the earlier papers.

Figure 2: I suggest the authors reverse the order of intensity (ie have sigma solid and black, and have the power a lighter line). I also suggest adding the 6 dB threshold to the second row.

Figure 8c: I think that Martos's data does need to be considered from the POV of the input magnetic track lines, especially from this sort of comparison.

Please also note the supplement to this comment:
https://www.the-cryosphere-discuss.net/tc-2018-53/tc-2018-53-RC2-supplement.zip
* * *

---

## Author Comment (AC1) · 22 Jun 2018

**Author response by Tom Jordan on behalf of all other authors, June 2018**

First, we would like to thank both reviewers for their constructive and thoughtful reviews and the editorial team for handling our submission. We have been able to fully address the suggested changes and we think that our manuscript is improved from the previous submission. Our responses and actions are in blue text.

**Review 1**

**General Impression:**

Jordan et al. analyze an extensive radar dataset covering large parts of Greenland. They introduce a novel metric ("the bed reflectivity variability") to locate basal water at the ice-bed interface. This is potentially a sufficient (but not a necessary) condition for water at the ice sheet base. The results can, for example, be used as a constraint for modelling the evolution of the Greenland Ice Sheet, which critically depends on the conditions at the basal boundary (wet vs. dry).

I have read this paper with great interest, and the authors do an excellent job in guiding the reader through the manuscript. Basic methodology is explained succinctly (incl. appropriate references), and novel parts are correspondingly highlighted and more detailed. All Figures are informative and of high quality. I am impressed with the scope of the analysis, which includes a very rich radar dataset, and the careful interpretation of the "basal reflectivity variability" as a new (but not the only) metric for basal water. In the following, I have a number of comments which should be addressed, and which hopefully will make the paper stronger. I believe that this paper will be useful for other researchers in the future, and apologize to the authors, and to the editor, for the delayed submission of this review.

Kind regards, Reinhard Drews, Geosciences, University of Tübingen, Germany

Thanks – this is a really tight summary of both the purpose and results of our study.

**Interpretation of basal reflectivity variability**

The authors make it clear (in I. 368) that reflectivity variability above the chosen threshold is a sufficient criterion for basal water. However, the converse argument (that is: low reflectivity variability, hence, absence of basal water) does not hold. It is very important that people using the results of this study are aware of this, and it should be mentioned more clearly (i.e. in the abstract and elsewhere). The importance of this point is highlighted because the authors themselves misinterpret their own results in this regard. The inference that "...basal water is often concentrated in the fast-flow onset regions and tributaries whilst it is **absent** from the main trunk." (I. 457) is not valid, because the absence of elevated bed reflectivity does not necessitate the absence of basal water (it could simply be homogenously distributed and thus not be visible with this metric). I see much potential for misinterpretation here, and the authors should describe the limitations of this novel metric in a more pronounced way.

We agree - it is vital that our basal water criterion is interpreted correctly (particularly by nonradioglaciology specialists who may be interested in the results but not the fine details). In light of this comment we have made the following changes:

- (i) We have been upfront in 4 key sections (abstract, final paragraph of intro, intro to results, conclusion) that the RES diagnostic is a `sufficient but not necessary' criteria for basal water.
- (ii) We have change title of section 2.5 to: *Interpretation of reflectivity variability as a sufficient diagnostic for basal water*
- (iii) We have deleted the section related to L 457, where we (likely mistakenly) interpret the absence of basal water.
- (iv) In Sect. 4.6 we now cross-reference the basal reflectivity/water map in Chu et al. 2018. This provides further support to our assertion that water diagnostic is well tuned to capture finite water bodes in the interior/fast flow initiation region of Petermann but will not capture homogeneously reflective wet sediment in the main trunk.

**2. Derivation of basal reflectivity variability:**

Does it matter that only log-transformed variables are used? I wonder this, because:

 $VAR \ 10 \ log 10(X) \neq 10 \ log 10 \ VAR \ X$

I can see the convenience of the log-transform when interpreting the reflection amplitudes, but I am unsure if this causes problems when calculating variances (and variance of multiplicative variables). Is there an underlying assumption about the statistics/uncertainties that was not explicitly mentioned? I understand this is a somewhat diffuse comment.

**This is a helpful comment. See our related response to reviewer 2.**

**3. Crossing-Over Analysis of the bed reflectivity variability**

The authors should do a crossing-over analysis of the bed reflectivity variability. This would strengthen the argument that birefringence and radar system specifics are small. It will also more clearly demonstrate the robustness of the new metric and highlight its advantages (which is that radar data collected over multiple field seasons and with various radar systems can efficiently be combined).

**We have added an Appendix A demonstrating cross-over statistics using MCoRDS v2 (the most recent and spatially extensive set of measurements) as a baseline.**

When interpreting this analysis, it is important to note that the cross-over standard deviations for sigma\_[R] are not necessarily to be interpreted as standard errors. This is because the flight-track windows that are compared, do not sample the same region of the glacier bed (and are likely significant overestimates). This contrasts with performing cross-over analysis of bed-echo power/reflectivity where the Fresnel zone defines a spatial overlap. As a form of edge detector, the purpose of the sigma\_[R]

metric is to identify a signal attributable to basal water within a 5 km region (rather than to coarse-grain an average value at the window size). Additionally, the along-track data in Fig. 2 shows that sigma\_[R] can rapidly fluctuate at a 5 km length scale. We therefore point toward the high degree of spatial structure in the water predictions (including intersecting flight lines from multiple years) as evidence for the robustness of the approach.

**4. Discrimination between water patches and smooth--rough transitions**

Variability in bed roughness is a competing mechanism which would also result in elevated bedreflectivity variability. The authors are aware of this and suggest that the threshold is well tuned to discriminate between these two scenarios. However, I did not fully understand why this is the case and the manuscript is in this regard unfortunately too vague (e.g. "...we demonstrate later ..." (I. 309) but where is this actually done?). This is important, because interpretation of the "bed reflectivity variability" as a proxy for basal water is a central conclusion of this paper, and other options must be convincingly excluded.

We agree with the reviewer that we need to be clearer on this point. In previous work, Jordan et al. 2017, we surveyed radar bed-roughness in northern Greenland (using both topographic-scale roughness, and the bed-echo abruptness/peakiness as a proxy for radar wavelength scale roughness). A visual comparison between the water map in this study and the roughness maps in Jordan et al. 2017, demonstrates that the water hits can occur in both smooth and rough regions of the bed. Moreover, comparison with ice-sheet scale spectral analysis of roughness by Rippin 2013, also demonstrates that both relatively rough and smooth regions can have water present (at least the length-scale considered).

In summary, we therefore think that the water diagnostic is not significantly modulated by spatial patterns in bed roughness, and is therefore is well tuned for ice-sheet scale analysis (by contrast, the previous water detection method used in Greenland by Oswald and Goginneni 2008, 2012 will only work in spatially-extended smooth regions where there are peaky waveforms).

We now realise that this is better incorporated in the discussion so have:

- (i) removed L. 309 from the methods.
- (ii) In sect 4.5 been more explicit about the lack of apparent modulation by roughness of our water map.

Additionally, we also now note in Sect 2.5 that whilst roughness (specular/diffuse scattering) transitions likely correlate with wet-dry transitions, this will actually act to amplify the signal component we are interested in (high values of sigma\_[R])

Finally, cross-over analysis of sigma\_[R] for the lower frequency (150 MHz) radar systems against the 195 MHz (Mcords v2) benchmark, does not show a significant bias (see the new Appendix A). This is further supporting evidence for (lack of) power modulation due to roughness and the assumptions made in the extraction of bed echo power (i.e. we aimed to suppress roughness-induced variability by integrating bed-echo power in fast time over the echo envelope). This has been added to 2.5.

**5. Temperature profile near ice divides**

I disagree that in the interior of ice sheets GHF and vertical diffusion are the dominant processes. What about vertical advection? In Greenland the surface mass balance is significant in the interior and thus I would expect a (strongly) non-linear temperature profile with depth (which would not be the case if only diffusion was important). A quick-look at the NGRIP profile does confirm this. Can you comment on this?

We agree with the reviewer. We have now added `and vertical advection' to line 393.

**6. Attenuation correction using modelled temperatures**

Briefly explain what the modelled temperature field of Goelzer et al., 2013 is based on, (how it compares to in-situ measurements), and how it may impact your results. Is it possible that temperature variations near the coast (where ice streaming is significant) are smoothed out, and thus do not correctly cancel the attenuation in your approach?

We have now added the following text about the temperature field to Sect. 2.2

The temperature field derives from a full 3D thermomechanical simulation over several glacialinterglacial cycles and is subsequently rescaled to a 1 km representation of the Bamber et al. 2013 ice thickness data set. The geothermal heat flux in GISM was initially taken from Shapiro and Ritzwoller 2004, but further adjusted with Gaussian functions around the deep ice core sites to match observed basal temperatures. Vertical temperature profiles are within 1-2 degrees C when compared to available in-situ measurements. GISM resolves the flow on a model resolution of 5 km, which causes some smoothing of the temperature field in narrow outlet glaciers near to the coast.

We therefore agree that the horizontal resolution of the temperature/attenuation field is coarser than some of the narrow flow features around the ice-sheet margins. However, our `sensitivity/perturbation analysis' in Sect 2.6 does provides a way to assess sources of bias in the attenuation model (including temperature). Note; we take a conservative approach of eliminating water-hits that do not satisfy the sigma\_[R]>6 dB criteria when the model is perturbed.

**7. Ice Fabric Variations**

I agree that fabric variations are a small component in the overall backscattered power budget. However, could it be that the "corridor" along the central ice divide is to some degree linked to ice anisotropy? Across a divide, ice fabric can change abruptly (compared to your 5 km window) (e.g. Martin et al., J. Geophys. Res., 2009; Drews et al., J. Glac., 2012) and potentially this may impact the inferred "bed reflectivity variability".

This is an interesting point! We interpret this as corresponding to regions where there is a pronounced azimuthal shift in the dielectric principal axes at a scale < 5 km. However, without access to polarimetric sounding data, this is difficult to test (and, in the general case where there are no cross-overs, correct

for). Whist we acknowledge the role of power modulation due to birefringent propagation (e.g. Matsuoka et al. 2012c), we do not think that flight track orientation (which would relate to the proposed fabric mechanism) has a dominant influence/bias upon the water hits. Supporting evidence for this, is the fact that the water hits in the ice-divide regions - see zooms 7(b), 7(d), 9(d) - occur a range of flight orientations relative to the ice-divide (which should correlate with the orientation of the dielectric principle axes).

We have added this point to the discussion (Sect. 4.6) along with referencing Martin et al .2009, Drews et al. 2012 and Matsuoka et al. 2012 cfor extra context upon the impact of ice fabric.

As an aside, power modulation (or lack of) due to ice fabric is another potential reason why we believe reflectivity variability to have certain `calibration advantages' over mapping basal reflectivity (which combines reflection values from multiple orientations). This is because, if we consider the (likely more general) case where the dielectric principal axes vary negligibly in orientation over a 5 km linear window, then there will be less potential power modulation due to fabric than if one were combining measurements from multiple orientations (e.g. Fig. 7 in Matsuoka et al. 2012c, JGR).

**Minor Suggestions**

Maybe "A sufficient constraint upon basal water distribution beneath the Greenland Ice Sheet from radar bed echo variability" would be a better (and slightly shorter) title?

If possible, we would like to retain 'A constraint upon the basal water distribution and thermal state of the GrIS from radar bed-echoes' as the connection with the thermal state/basal temperature is a central purpose of the MS (note -we have now dropped 'basal' from basal thermal state to shorten the title) However, we have now been explicit in the abstract that our water diagnostic is interpreted as a sufficient (but not a necessary) criteria for basal water.

I. 320 There is a missing section number in the internal reference.

We have now added Sect. 2.2.

I. 580 Basal 'freeze-on' is one, but not the only explanation for the disturbances seen at larger depth. Concerns about this (e.g. Dow et al., Geophys. Res. Lett., 2018) or other explanations (e.g. Bons et al., Nat. Comms., 2016) should also be mentioned here.

A good point. We have now referred to the englacial features using the more generic term `Basal units of disrupted radiostraitigraphy' (following Dow et al. 2018) and added the additional explanations in Bons et al. 2016 (anisotropic rheology), Wolovick et al. 2014 (stick-slip mechanism).

Fig. 5b add y-label ([R] (db))

Done

**Review 2**

Summary: This paper uses a RES diagnostic the author's term "bed-echo reflectivity variability" for the long archive of RES observations over Greenland to get at the distribution of basal water. They then proceed to compare this to various prediction for the distribution of subglacial water. It is comprehensive and thorough, however I cant help thinking its being presented as a lot more sophisticated than it actually is. High passing radar data has been a (justifiable) refuge of radioglaciologists since the C-130 TUD days, and thats basically what seems to be happening here - just in slow time rather than fast time.

We do agree that, viewed from a technical standpoint, our radar method is (relatively) simple. However, the novelty and impact of our study is primarily due to our geographical analysis of an extensive radar data set across the ice sheet. This is the first time that an ice-sheet-wide assessment of basal water has been done with a post-2003 radar data set.

Major issues: Novelty: Bed echo variability has been long used for characterizing basal interface (Neal, 1982; Peters et al., 2005; Carter et al., 2007), and the surface interface (Grima et al., 2014) - most of this literature is not mentioned from this context. The authors here extend to very long length scales, and integrate in fast time over the echo to suppress roughness effects in an attempt to essentially map out subglacial water using an assumption of bimodal wet/dry distribution, to get around variability in attenuation that will inevitably bias absolute values.

We apologise for the previous omission of this literature, and now have included an extra paragraph to the relevant methods section (2.3) describing the prior work of Neal, Peters, Carter and Grima, clarifying the differences with our approach:

'It is important to clarify the difference between the use of bed-echo power/reflectvity variability in this study from previous radioglaciology studies (Neal1982, Peters et al. 2005, Carter et al. 2007, Grima et al. 2014). These studies focused upon the variability/statistics of the peak echo power as a result of phase modulation by interfacial roughness. By contrast, in this study we suppress roughness effects by integrating power in fast time over the echo envelope. We are therefore able to focus upon power variability that is a result of along-track changes in the bed dielectric.'

Edge detection: The approach I feel is misnamed. From Figure 2 it seems clear that large scale changes dominate their analysis, and thus basically what the authors have is an edge detector. What they are finding is not so much variability as gradients. In order to get at the small scale variability indicated in figure 5b, they would have to high pass the data, which they are not doing. A multi-scale approach may be more productive to get at mixed media cases.

The reviewer is correct and the approach we take can be viewed as a form of edge detection. However, the purpose of plot 5b is to show that the approach is not limited to a singular transition (i.e. the variability approach will be able detect more water hits than if a singular transition/conventional edge detector is imposed). In other words, we are not necessarily interested in detecting/classifying fine-scale variability, (we just did not want to impose that a singular dielectric transition was present).

It is important to bear in mind, that our method was tuned around the (in our view necessary requirement) of making a comparison to the basal thermal state synthesis by Macgregor et al. 2016 (this assumed a resolution of 5 km, which is also an appropriate scale for informing ice-sheet scale numerical models). This is specifically why we did not consider a multi-scale approach. Additionally, previously identified GL water bodies are small (< 5 km), so we were concerned that if we imposed a singular transition at the prescribed resolution, then we would miss a significant fraction of the basal water.

Based upon this helpful comment, we have made the following changes:

- (i) In the abstract, and other key sections where we discuss bed-echo reflectivity variability we have highlighted that it acts as a form of edge detector.
- (ii) We have been explicit that we used this approach (rather than other methods of edge detection) as we do not wish to limit ourselves to a singular wet-dry/dielectric transition in a 5 km window.
- (iii) We now have now revised the introduction to Section 2.3, better explaining the context for the introduction of our method (particularly, motivating our choice of length-scale regarding the comparison later made with Macgregor et al. 2016).

Statistics in dB space: I am concerned at the application of statistics in dB space, and think this needs to be better motivated. Due to the compression of the distribution of the echoes using the attenuation model, and the highly bimodal reflectivity of the bed, they 'get away with it' somewhat; however, I attach an jupyter notebook that attempts to illustrate the complexities of doing the variability statistics in dB using a synthetic fractal distribution (again, the hypothesized distribution will be more bimodal, but there will be a sensitivity long wave length errors in attenuation) and a bimodal distribution.

We thank the reviewer for their detailed feedback on this subtle point and have read their jupyter notebook with interest. We have now added the qualifying statement to Sect. 2.3 that we consider the variability of the log-transformed variables (and that this differs from the variability of the linear variables in log space). However, since our 6 dB water threshold was devised with the log-transformed variables in mind, we genuinely do not think that there is an issue with our approach (and, for reasons given below, we think that are certain benefits).

Specifically, our motivation for using dB space is:

- (i) That it enables a clearer connection to be made with the dB reflection amplitudes for various geophysical media (which are more familiar to the radioglaciology community than linear Fresnel values). It is also to be noted that the radioglaciology community (either implicitly or explicitly) apply reflectivity statistics in dB space regarding water detection (e.g. MacGregor et al 2013, Wolovick et al. 2013).
- (ii) The log-transform enables us to consider the additive form of the radar power equation (which results in the simpler to interpret VAR([R]) +VAR([L]) rather than VAR(R\*L))

These points have now been added to Sect 2.3, along with a qualifying statement in 2.6 that the 6dB threshold only applies to log-transformed reflectivity.

Calculation of sigma: It wasn't clear if you are taking the deviation in power of points separated by 5 km, or just taking the standard deviation of all points within the 5 km window.

We take the standard deviation of all points within the window. This has now been made clear in Sect 2.3.

Ice surface transmission losses: Surface losses due to roughness or near surface englacial water are not considered. I think for this paper they could be important, as they do not correlate with the predictions for whole ice sheet attenuation, and have the potential for sharp gradients. Surface and near surface losses should be addressed maybe just by a demonstration that they are negligible.

Regarding this point, it is important to bear in mind the central purpose of our study and the impact upon the central results/conclusions (specifically, the assessment of regions of basal thaw & the comparison we make with Macgregor et al. 2016 and the GHF maps). We note that regions susceptible to surface water/surface roughness-induced variability are toward the margins/faster flowing. We are therefore only likely to get `false positives' for basal water (i.e. anomalously high variability) in regions where there is a already a high degree of confidence that there is a warm thermal regime. This surface modulation therefore, does not have a large impact on the thermal state comparison (since we are primarily interested in water hits in the slow-flow regions previously predicted to be frozen), or the GHF comparison (since we limit this comparison to the interior ice divides).

In summary; whilst we do agree with the reviewer that these effects are present, we believe that there will be minor impact upon our take-home results/conclusions. However, we do agree that this should be added to the discussion and have now added an extra paragraph. And referenced Grima et al. 2014 and Schroeder et al. 2016a regarding surface roughness—induced power variability (see Sect 4.6).

**Minor issues:**

Data traceability: The authors need to emphases that these data are from the CSARP processor found on the KU website, and NOT the MDVR processed data found on the NSIDC website. The latter should not be used for quantitative analysis of the bed echo. NSIDC says this in their website, but a disclaimer here might help head off confusion.

In sect 2.1 & and the data availability section we have now added explicitly that we are using the CSARP data.

Along track processing: There is very little detail on how azimuth processing has changed over time, and how this could effect the results.

The new Appendix A (cross-over statistics for reflectivity variability) provides an instrument-byinstrument breakdown, demonstrating no significant/minor cross-over biases. This provides an empirical test that the data is suitable for combined interpretation. We already give 3 references in 2.1 regarding radar signal processing (Rodriguez-Morales et al 2014, Gogineni et al 2014, Paden2015), and believe that presentation is at the correct level of detail for the (primarily glaciological) readership of TC.

**Also not clear: did the author do along track incoherent averaging as they did for earlier papers?**

No – this was not done. This was discussed with the CReSIS radar team, and was deemed an unnecessary (but inconsequential) step in the prior work, with the (non-averaged) L1B data product being preferable. The overall effect of the prior-averaging is to decrease the spatial resolution. However, since in Jordan et al. 2016, we chose to grid the data anyway. there are no significant consequences for the maps in the paper.

We now have added that we did not do this step to Sect 2.1.

Power determination approach: The method for extracting aggregate power, and rejecting bad echoes, appears to have changed between Jordan et al 2016, 2017. In those works, a symmetrical window about the peak is taken, while in this work, a 10 dB threshold down from the peak power is used. A 10 dB SNR is used here, while in earlier work it appears to be a 17 dB threshold is used. The change needs to be better justified, and the opportunity taken to explain if there is any impact of the results of the earlier papers.

The reviewer is correct and we have applied a less strict SNR criteria in this study (10 dB rather 17 dB). The justification for this change is that we now believe that were overly strict in the previous studies. If the old 17 dB threshold is used then there is a decrease in the effective coverage in southern Greenland.

The impact upon the previous works is that the `effective coverage' of the radar flight-track data that is analyzed will be slightly smaller than for this paper. As the previous papers were technique (rather than data set) orientated papers we do not foresee any significant issues.

Figure 2: I suggest the authors reverse the order of intensity (i.e. have sigma solid and black, and have the power a lighter line). I also suggest adding the 6 dB threshold to the second row.

If possible, we would like to keep the sigma variables in color. However, we have now increased the line thickness of the sigma variables and decreased the line thickness/intensity of the solid black lines. The 6 dB threshold has also now been added in as suggested.

Figure 8c: I think that Martos's data does need to be considered from the POV of the input magnetic track lines, especially from this sort of comparison.

First, we interpret `this sort of comparison' in relation to the different spatial scales at which the radar water predictions and GHF models are assessed at and POV to mean `point of view'. In which case, we agree that it is important to add a qualifying section on this and have added:

In the comparison between the radar water predictions and GHF in Fig. 8 it is important to bear in mind that the GHF distributions are evaluated at a lower spatial resolution. For example, the resolution of the GHF distribution by Martos et al. 2018/in revision is a consequence of the spectral method (window size and overlap) which has an effective resolution of ~ 75 km.

Martos et al. 2018/in revision, is close to being accepted for publication at GRL and we hope to reference the published work in our final paper. The question of magnetic track spacing was dealt with extensively in their review process/paper and we therefore do not think it is necessary to include any more information in our MS. However, for completeness, we now briefly summarize the key points:

- Martos et al. 2018/in revision used magnetic anomaly data from the World Digital Magnetic Anomaly Map v2 (WDMAM v2) compilation to derive Curie Depths from which the GHF is estimated.
- The WDMAM v2 is based on a good line coverage of Greenland with datasets described in the WDMAM v2 report. The provided original format of these datasets contain grid cells between 1 and 5km. The sparsest line spacing separation in this compilation in a specific part of North Greenland with 60 km between track lines. The coverage is much denser over the rest of Greenland, reaching line spacings <10 km.</li>
- The (de-fractal) spectral method applied to the magnetic data uses window sizes of 350 km x 350 km with a 57% overlap. With the spectral method the depth to the top and the depth to the centroid of the deepest magnetic sources are identified and assigned to the central part of the window (spatial resolution of the spectral method would be ~75 km). These sources are, by definition, wide and present long wavelengths in the magnetic signal. These long wavelengths are by well resolved with this line spacing configuration already mentioned above.

**Additional changes**

Since the initial submission, Rutishauser et al. 2017 (AGU abstract) has now been published as a journal paper, so we have added the full reference.

We have also added two extra references for: (i) context on GHF/basal water comparisons (Siegert and Dowdeswell 1996), (ii) Ground water (Siegert et al 2017).

We have changed the reference in the flow-routing methods section to Wang and Li 2006.

---

## Author Response (AR2)

**Comments to editor corrections – Tom Jordan on behalf of all authors**

We would like to thank Kenny Matsuoka for his supportive and helpful editorial comments. Our responses are in blue text.

Editor Decision: Publish subject to technical corrections (07 Aug 2018) by Kenny Matsuoka

Comments to the Author:

The revised manuscript addressed reviewers' concerns adequately and is much stronger than the original version. Below, I provide a few specific comments to improve this manuscript further. Because all of them are minor, I am happy to accept this manuscript subject to technical corrections. If the authors intend to revise the manuscript more substantially or in a different way from the guidance below, please contact me prior to the submission of the final manuscript.

- The 6-dB criterion used in this manuscript is supported by the phase diagram presented in Fig. 5. The authors show that this criterion satisfies (and is slightly stricter than) a classical criterion of the 10-dB reflectivity contrast in most cases. They also emphasize that the map shown in this study is a conservative estimate. In this sense, the paper is complete. However, ideally speaking, it is better if the sensitivity of geographical distribution of thawed bed is discussed for slightly different criteria. If the authors used slightly different criteria (e.g. variability > 5 dB) in their initial work and get similar spatial patterns, please mention it in this manuscript. If such results are not available, please ignore this comment.

This is a helpful comment. We would like to highlight that Fig 3c - the spatial distribution of sigma_[R] - effectively contains this information (e.g. the light blue and dark blue regions indicate sigma_[R] < 4 dB). Moreover, the plot demonstrates that the spatially contiguous regions of low sigma_[R] (i.e. no evidence for water) are largely preserved from the 6 dB threshold. We believe that this provides additional evidence for the robustness of threshold choice (in the sense that similar spatial patterns are present).

We have therefore added a sentence on this when we introduce the sigma_[R] threshold (Sect. 2.6).

- Please reconsider to call "large" bed echoes as "good quality" bed echoes. I understand authors' intention to use a non-technical term but smaller bed echoes can reflect just the nature of the dry bed and can be in a good quality. I think that "bed echoes with high signal-to-noise ratio (S/N)" is a clearer way to call it, and it should be defined in such a way at least first time it appears.

Thanks for the guidance on this – we have now properly defined use of `good' and `poor' quality in Sect 2.1 in terms of SNR.

- slush "/" is over-used in my opinion. In many cases, they can be replaced with "and", which makes sentences clearer.

We have edited this throughout where we deemed appropriate (see marked version of the MS).

- L204-209: Spatial resolution of GISM can be mentioned earlier, at the line 204 (in the marked manuscript). The modelled ice thickness is different from the observed thickness so that the vertical variation of ice temperature is normalized to the modeled ice thickness, and then re-scaled to the observed ice thickness, I think. Statements at lines 206-208 are unclear for me, because it mentioned both this vertical scaling and probably the horizontal resolution adjustments between the 5-km-resolution model and 1-km-resolution DEM.

This interpretation is correct and thanks for pointing out the ambiguity. We have switched the paragraph ordering and added the spatial resolution of GISM earlier in the paragraph. We have also added the extra sentence: *In the calculation of <N> the temperature field was firstly interpolated to a 1 km resolution then vertically scaled using the 1 km representation of the Bamber et al. 2013 ice thickness.*

- L203: these -> those?

We could not see `these' on L 203. We checked other usages of `these' and could not find an error.

- L391: You said "initial results" at L369 and that only green and red data are analyzed further in Fig. 6 caption. Please mention this point in the main text as well for completeness.

This already in the main text (L 388 of the previous marked-up version). To better highlight what initial results mean we have added `pre- sensitivity analysis' to L369. We have also added an extra sentence when introducing the results, stating that the `red circles in results correspond to the set of red and green points in the sensitivity analysis'. Hopefully this is now clearer than before.

- L396: original paper? Replace with the full reference.

We have replaced with Jordan et al. 2016.

- L469-470: incomplete sentence?

Thanks for spotting this (we mistakenly deleted some text from the TCD paper). The full paragraph (same as TCD paper) reads*: A summary of local GHF estimates using borehole temperature profiles and thermomechanical model inversions (Weertman,1968; Dahl-Jensen et al., 1998, 2003; Greve, 2005; Buchardt and Dahl-Jensen, 2007; Petrunin et al., 2013) are provided by Rezvanbehbahani et al. (2017); Martos et al. (2018), and demonstrate general consistency between Fig. 8(c) and local estimates at GRIP, NEEM, NorthGRIP and Camp Century. Local estimates of GHF at Dye 3 (_ 20-25 mW m/2) are significantly lower than all three GHF models.*

- L515: spelling (space), f"_"ast.

Done

- L593: remove "the".

Done

- L598 – L604: so what's your point?

Thanks – we agree this now seems out of context as we had deleted some text from the TCD paper. The point we are making is that the comparison with ice velocity is likely to be limited in utility, since the water desitbrution is largely from pre-surface melt data. We have made this clearer by changing the final sentence to: *It is therefore essential to re-emphasise that the basal water predictions generally correspond to the winter storage (pre- surface melt) configuration and may, therefore, be of limited utility in understanding spatio-temporal patterns related to ice dynamics.*

- Fig. 1C: deep ice core sites are shown in blue in this figure but in yellow in Figs. 6-10. It is better if you show the core sites all in yellow.

A good spot. We have changed the core sites to yellow.

- Figure 2a second panel from the top: the threshold red bar is displayed for sigma_R = 10 dB, not 6 dB.

We have double-checked this point and the threshold line is correct (i.e. it is consistent with sigma_R = 6 dB).  To make this clearer/avoid confusion we have added to the caption that the dashed pink line refers only the *right axis.*

Thank you for submitting the valuable work to The Cryosphere.

Kenny Matsuoka
TC/TCD Editor

**Additional changes**

We have now added the full reference for Martos et al. 2018 which was recently published in GRL.

[revised manuscript text omitted]

*We have shortened the label of the 2nd column to make this suitable for a single column width.*

[revised manuscript text omitted]